

# Classification of iron oxide aerosols by a single particle soot photometer using supervised machine learning

Kara D. Lamb[1,2]

[1]Cooperative Institute for Research in Environmental Sciences, University of Colorado Boulder, Boulder, CO, USA
[2]NOAA Earth System Research Laboratory Chemical Sciences Division, Boulder, CO, USA

**Correspondence:** Kara Lamb (kara.lamb@noaa.gov)

**Abstract.** Single particle soot photometers (SP2) use laser-induced incandescence to detect aerosols on a single particle basis. Both refractory black carbon (rBC) and other light absorbing metallic aerosols, including iron oxides ($FeO_x$), have been characterized by the SP2, but single particles cannot be unambiguously identified from their incandescent peak height (a function of particle mass) and color ratio (a measure of blackbody temperature) alone. Machine learning offers a promising

approach for improving the classification of these aerosols. Here we explore the advantages and limitations of classifying single particle signals obtained with the SP2 using a supervised learning algorithm. Laboratory samples of different aerosols that incandesce in the SP2 (fullerene soot, mineral dust, volcanic ash, coal fly ash, $Fe_2O_3$, and $Fe_3O_4$) were used to train a random forest algorithm. The trained algorithm was then applied to test data sets of laboratory samples and atmospheric aerosols. This method provides a systematic approach for classifying incandescent aerosols by providing a score, or conditional probability,

that a particle is likely to belong to a particular aerosol class (rBC, $FeO_x$, etc.) given its observed single-particle features. We consider two alternative approaches for identifying aerosols in mixed populations: one with specific class labels for each species sampled, and one with three broader classes for aerosols with similar properties. While the specific class approach performs well for rBC and $Fe_3O_4$ ($\geq$99% of these aerosols are correctly identified), its classification of other aerosol types is significantly worse (only 47-66% of other particles are correctly identified). Using the broader class approach, we find a

classification accuracy of 99% for $FeO_x$ samples measured in the laboratory. The method allows for classification of $FeO_x$ as anthropogenic or dust-like for aerosols with effective spherical diameters from 170 to >1200 nm. The misidentification of both dust-like aerosols and rBC as anthropogenic $FeO_x$ is small, with $< 3\%$ of the dust-like aerosols and $< 0.1\%$ of rBC misidentified as $FeO_x$ for the broader class case. When applying this method to atmospheric observations taken in Boulder, CO, a clear mode consistent with $FeO_x$ was observed, distinct from dust-like aerosols.





# 1   Introduction

The single particle soot photometer (SP2) has been used over the past decade to quantify refractory black carbon (rBC) mass and internal mixing on a single particle basis (Stephens et al., 2003; Schwarz et al., 2006). Recently, the SP2 has been increasingly used to quantify other light absorbing refractory aerosols (e.g. Moteki et al. (2017); Liu et al. (2018)). In particular,

observations in source regions have shown that iron oxide containing aerosols from anthropogenic origins are present in the atmosphere (Liati et al., 2015; Dall'Osto et al., 2016; Adachi et al., 2016; Li et al., 2017), and these aerosols can be detected via laser-induced incandescence with an SP2 (Yoshida et al., 2016; Moteki et al., 2017). These aerosols have been found to be mostly pure iron oxides that are fractal aggregates of $\sim$100 nm spheroids, internally mixed (heterogeneously) with nitrogen or sulfate (Dall'Osto et al., 2016; Adachi et al., 2016; Li et al., 2017). They have been linked to transportation sources (engine

exhaust, traffic brake wear) and industrial sources such as steel processing (Ohata et al., 2018). Iron oxide aerosols quantified by the SP2 were referred to as $FeO_x$ in past literature (e.g. Moteki et al. (2017)), and we continue this convention here. In general, $FeO_x$ as quantified by the SP2 in atmospheric observations in past studies potentially included aerosols from both anthropogenic and non-anthropogenic sources. The mass mixing ratio and size distribution of $FeO_x$ has been quantified in East Asia, where observations suggested these aerosols were mainly from anthropogenic sources, and were also observed to

be significantly more prevalent than previously believed (Yoshida et al., 2016; Moteki et al., 2017; Ohata et al., 2018; Yoshida et al., 2018). These measurements have important implications for the climatic effects associated with these aerosols: the direct radiative climate effects of anthropogenic $FeO_x$ may be as important as brown carbon in some regions (Moteki et al., 2017; Matsui et al., 2018), and modeling studies based on these measurements indicate these aerosols could also be an important source of particulate iron for the oceanic biogeochemical cycle (Matsui et al., 2018; Ito et al., 2018).

Improving the detection of iron oxide aerosols linked to anthropogenic sources is key to understanding their potential impact on the climate. The SP2 offers a promising method for real-time quantification of these aerosols, as previous detection techniques are limited to off-line methods such as X-ray spectrometry (Adachi et al., 2016). However, while laser-induced incandescence can be used to quantify the mass of pure magnetite ($Fe_3O_4$) and to a lesser extent, hematite ($Fe_2O_3$) and wüstite (FeO) (Yoshida et al., 2016), the interpretation of ambient SP2 observations has been limited by the misclassification of other

aerosols as $FeO_x$, including both rBC and aerosols containing metallic components from non-anthropogenic sources.

To first order, $FeO_x$ can be differentiated from refractory black carbon (rBC) because of differences between the blackbody temperature and peak incandescent signal (relative to the particle mass) associated with single particles incandescing in the laser of the SP2. However, the temperature and incandescent peak height alone are not sufficient to unambiguously identify $FeO_x$. Because $FeO_x$ has a higher mass to incandescent signal relationship than rBC (Yoshida et al., 2016), and is generally

significantly rarer than rBC in the atmosphere in a source region by a factor of $\sim$250x (Moteki et al., 2017), the misclassification of even a small fraction of rBC as $FeO_x$ can bias the retrieved mass mixing ratio.

In addition, other types of metallic aerosols (e.g. tungsten, silicon, chromium, niobium, gold, and aluminum) can be detected via laser-induced incandescence (Stephens et al., 2003; Schwarz et al., 2006). Although these aerosols are unlikely to be significantly present in the atmosphere, more common aerosol classes, such as coal fly ash, mineral dust, and volcanic ash also



can contain metallic inclusions that are detected with low efficiency by the SP2, and in some cases have similar blackbody temperatures/incandescent peak heights as FeO$_x$. Laboratory tests performed for this study on different samples of fly ash (Miami F, Welsh C, and Clifty-F), a coal combustion product that is a significant source of metallic particles to the atmosphere, indicated these aerosols incandesce with low efficiency (i.e. only a small fraction of the particles have a non-zero incandescent

signal in the SP2). This low detection efficiency likely indicates that only a small fraction of these aerosols contain sufficient quantities of materials that can be heated to detectable incandescence in the SP2 laser. Approximately 5-10% of volcanic ash particles from the Eyjafjallajökull volcano incandesced in the SP2 during the SOOT 11 campaign, with greater incidence of incandescence for larger particles (Heimerl et al., 2012). Incandescent particles in Icelandic mineral dust and Taklamakan Desert dust also have been detected with low efficiency using the SP2 (Yoshida et al., 2016). An SP2 was used in one study to

estimate the hematite content in Saharan dust measured off the West coast of Africa, and demonstrated good closure between the hematite concentration associated with single dust particles and the optical properties observed in the dust plume (Liu et al., 2018). While previous work focusing on anthropogenic FeO$_x$ has relied on using the optical size of these aerosols after any volatile coatings have evaporated as an additional criteria to differentiate anthropogenic and non-anthropogenic aerosols with metallic components (Yoshida et al., 2016; Moteki et al., 2017; Ohata et al., 2018; Yoshida et al., 2018), this method is limited

to the range over which the SP2 optically sizes these aerosols (generally limited to ∼170-350 nm volume equivalent diameter for FeO$_x$). Previous studies using the SP2 to quantify FeO$_x$ associated with anthropogenic sources have also not provided quantitive measures of classification performance for these aerosols.

Here we demonstrate that supervised machine learning can be used to differentiate laboratory samples of pure FeO$_x$ from other types of incandescent aerosols expected in the ambient. Machine learning refers to a number of related algorithms using

optimization techniques based in probability theory to directly extract information from observations, without relying on *a priori* knowledge of underlying physical models. Supervised machine learning methods, which use labeled data sets to initially train algorithms, are particularly suited to classification problems. These methods are used e.g. to classify images, for text-to-speech applications, and for identifying handwritten digits, and are also increasingly being applied to scientific applications, including atmospheric aerosol measurements. While machine learning approaches have been used to classify single particle

aerosol mass spectra (Zawadowicz et al., 2017; Christopoulos et al., 2018) and biological aerosols detected via ultraviolet light-induced fluorescence (Robinson et al., 2013; Ruske et al., 2017, 2018; Savage and Huffman, 2018), they have not yet been applied to the problem of classifying aerosols detected via laser-induced incandescence. We review detection of incandescing aerosols with the SP2 and describe measurements on laboratory samples in Section 2. We discuss how features derived from single particle signals can be used as input to a supervised learning algorithm and describe a method for training and optimizing

this algorithm in Section 3. In Section 4, we discuss the performance of the trained random forest algorithm on laboratory samples and atmospheric observations. This method extends the classification of FeO$_x$ associated with anthropogenic sources beyond the range over which the SP2 can optically size these aerosols and also reduces the misclassification of other aerosols as anthropogenic FeO$_x$.



## 2  SP2 detection of incandescing aerosols

To optimize classification of aerosols measured by the SP2, we first describe the detector configuration, calibrations, and laboratory measurements used in this study and discuss how different aerosols are detected by the SP2. SP2s operate by using laser-induced incandesce (L-II) to detect sub-micron incandescing aerosols on a single-particle basis (Stephens et al.,

2003). Their operation has been discussed in detail elsewhere (Schwarz et al., 2006, 2010; Moteki and Kondo, 2010). The SP2 determines the mass of the incandescent portion of single aerosol particles by using an ND:YAG laser (1064 nm) to heat refractory particles with a sufficient absorption cross-section to vaporization. Aerosol particles are observed by four detectors as they traverse the laser beam, with two detectors measuring the incandescent signal in the visible, and two measuring scattered light at 1064 nm. For the study, we define "incandescent" aerosols as those that have a non-zero signal in the two incandescent

channels.

### 2.1   NOAA SP2 detector configuration

The two incandescent detectors measure light emitted from the particles in distinct wavelength bands, providing a measure of the spectral dependence of incandescence, which can be converted to a temperature (Moteki and Kondo, 2010). For this study we use a customized SP2 (the NOAA SP2) whose detector configuration differs slightly from the commercial versions

(Droplet Measurement Technology, Longmont, CO), and which was previously described in Schwarz et al. (2006, 2010). This SP2 is operated with 4 detector channels and a 5 MHz acquisition rate. In the typical configuration of the NOAA SP2, a "red" incandescent detector is a photomultiplier tube (PMT) with a peak sensitivity at 630 nm (450-650 nm, Hamamatsu H6779-20) and a "blue" detector is a PMT with peak sensitivity at 420 nm (350-450 nm, Hamamatsu H6779). (Alternatively a PMT (Hamamatsu H6779-02) with a peak sensitivity of 500 nm and a smaller range (450-630 nm, "orange" detector) is sometimes

used in place of the red detector in the NOAA SP2.) An additional Schott glass band-pass filter (KG5, 330-665 nm) in front of the red (orange) detector removes light at wavelengths longer than 750 nm to avoid detection of scattered pump laser light (at 807 nm, see Figure 1 for detector sensitivity ranges). The NOAA SP2 also uses a 2x1 mm aperture and a short-wave pass filter (SWP-730, Spectrogon) in front of the red detector to further reduce sensitivity to scattered light from the pump laser. Color temperature ratio is calculated from the ratio (blue:red or blue:orange) of the measured signals at the peak of incandescence

(in practice, an average over a small range around each peak is used to reduce sensitivity to high frequency noise). In this work, the gains on the SP2 blue and red channels were chosen so that the distribution of the color ratios for ambient black carbon is centered near 1. Due to a shift towards the red of blackbody radiation for cooler objects (See Fig. 1, left panel), the characteristic boiling temperature for iron oxide aerosols (∼3300 K, vs. ∼4320 K for rBC) corresponds to a color temperature ratio of ∼0.7, relative to rBC at 1.0 (Fig. 1, right panel).

The mass of the portion of the particle that incandesces can be determined from the peak height of either incandescent signal, which in the case of rBC is linearly proportional to its mass over most of the accumulation mode (Schwarz et al., 2006; Moteki and Kondo, 2010; Gysel et al., 2012). In this work, we use the blue incandescent peak amplitude to derive single particle incandescent mass, and show incandescent peak height (linearly) scaled based on the rBC mass calibration (as this provides a





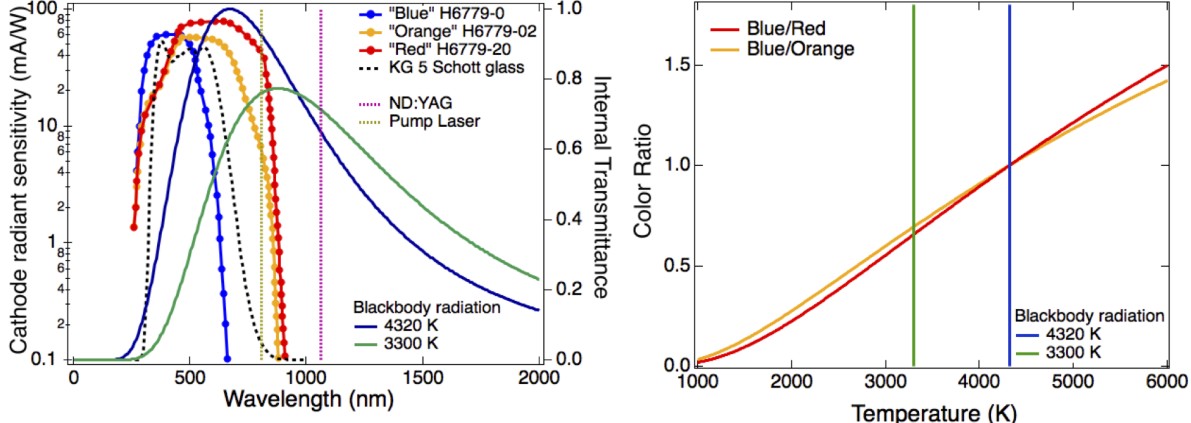

**Figure 1. Determination of single particle blackbody temperature from SP2 incandescent detectors.** *Left:* Normalized blackbody curves for T=4320 K (typical of rBC) and T=3300 K (typical of FeO$_x$) are shown as solid blue and green curves, along with the cathode radiant sensitivity of the PMT's typically used in the SP2, as blue, orange, and red lines and markers. The dashed black line gives the transmissivity of the glass filter used with the red and orange detectors. The yellow dashed line is the wavelength of the pump light, and the maroon dashed line is the wavelength of the ND:YAG laser. *Right:* The color ratio as a function of the particle's characteristic blackbody temperature derived for two different detector configurations (blue:red, blue:orange) is shown, assuming the color ratio is scaled to 1 at 4320 K.

physical metric that is not dependent on the detector gain settings). The detection efficiency of FeO$_x$ is dependent on the SP2's laser power. Although magnetite can be detected with nearly 100% efficiency under typical conditions (Yoshida et al., 2016), the detection efficiency of hematite is lower and dependent on the particle's total hematite mass. Up to a point, higher laser power increases the efficiency with which the smaller FeO$_x$ aerosols can be detected, as the higher laser power compensates for

their smaller absorption cross-sections. The SP2 is insensitive to goethite and ferrihydrite, as their absorption cross-sections at the wavelength of the ND:YAG laser are not sufficient for these aerosols to be heated to incandescence (Yoshida et al., 2016).

Incandescing aerosols can be simultaneously optically sized using the scattering channels in the SP2. The optical size is determined by an avalanche photodiode (APD) with sensitivity at 1064 nm (model C30916E, Perkin-Elmer Optoelectronics, Quebec, Canada). The SP2 additionally uses a position sensitive detector (a four quadrant silicon APD, Perkin-Ellmer

C30927E-01) to determine the position of the particles in the beam with respect to the center of the laser as has been described in detail in Gao et al. (2007). The SP2 used in this study can be run with either a high gain scattering channel setting or a low gain scattering channel setting. The high gain setting (5x higher than the low gain setting) is optimized for detection of rBC in the accumulation mode (typically ∼90-550 nm). The low gain setting allows the SP2 to optically size larger aerosols, although a significant fraction of the non-rBC materials cannot be optically sized even with this lower gain setting. The measurements

used in this study use the high gain setting as these are the typical settings used during past aircraft campaigns.





**Table 1. Laboratory aerosol samples used in analysis.** We test different aerosols with known incandescent components, given in the table below. The sample size refers to the total number of incandescing aerosols in the sample.

| Material | Sampling Method | Sample size |
|---|---|---|
| Fullerene Soot | Nebulizer | 231,101 |
| Fullerene Soot + Glycerol | Nebulizer | 162,959 |
| $Fe_3O_4$ powder (<5$\mu$m) | Nebulizer | 258,624 |
| $Fe_2O_3$ powder (<5$\mu$m) | Nebulizer | 45,609 |
| Clifty-F Fly Ash | Nebulizer | 18,677 |
| Arizona Test Dust | Nebulizer | 67,102 |
| Volcanic Ash | Nebulizer | 33,970 |

## 2.2 Calibrations

The laser power of the SP2 used in the analysis was calibrated with 220 nm polysterene latex spheres before and after each data set was taken (Schwarz et al., 2010). The incandescent signal to mass relationship for rBC was calibrated by using fullerene soot (Lot #F12SO11) size selected at different mobility diameters (between 150-350 nm) through a differential mobility analyzer

(DMA), along with the mass to mobility diameter relationship for rBC from Moteki and Kondo (2010). The incandescent to mass relationships for laboratory samples of both magnetite and hematite have previously been characterized (Yoshida et al., 2016), and we determine the $FeO_x$ mass relative to the rBC mass calibration using those relationships (See Figure 2b).

## 2.3 Preparation of laboratory data sets

A data set was compiled from laboratory samples to simulate aerosols expected to be found in the atmosphere. Data from labo-

ratory samples of fullerene soot (Lot #F12SO11), magnetite ($Fe_3O_4$, < 5 microns, Sigma Aldrich 310050), hematite ($Fe_2O_3$, <5 microns, Sigma Aldrich 310069), Arizona Test Dust (PTI ISO 12103-1), volcanic ash (VA) from the Eyjafjallajökull volcano (collected on the ground in Iceland), and coal fly ash (Clifty-F, referred to as FA) were measured in the laboratory. Fullerene soot is a calibration material that behaves in the SP2 similarly to ambient rBC (Kondo et al., 2011; Baumgardner et al., 2012). Arizona Test Dust (ATD) is a commonly used reference material for mineral dust and includes some metallic components,

including 2-5% by weight of hematite. Thickly coated rBC particles were simulated by mixing glycerol (99.5%) with fullerene soot. Each of these samples include some fraction of particles that incandesce in the SP2 laser, although ATD, VA, and FA only have a small fraction of incandescent particles relative to the particles that do not incandesce. Samples were measured with a flow rate of 4 cc/s and at low enough concentrations to avoid cases of two incandescent particles crossing the laser at the same time. Because machine learning algorithms perform best with a large number of examples (to provide sufficient variance), we

focused on acquiring a large number of measurements for each aerosol type. The total number of laboratory samples, which are subdivided between the training and test sets (discussed in the next section) are given in Table 1. The incandescent peak height



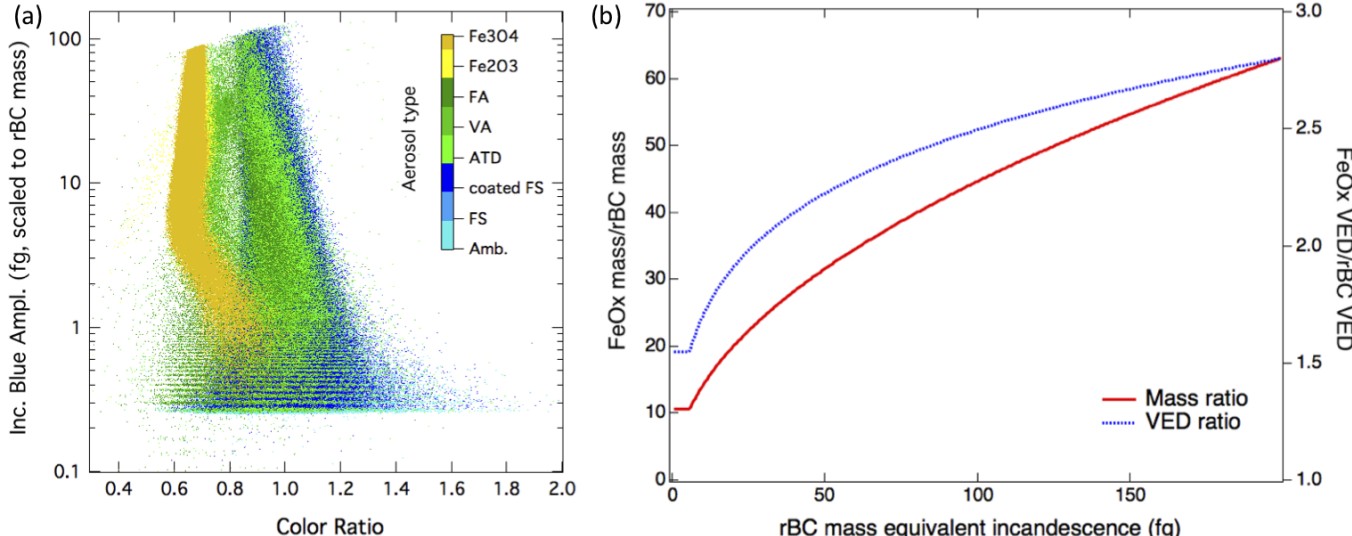

**Figure 2. (a) Incandescent peak height to color ratio relationship for different incandescent aerosol types.** Laboratory samples of different test materials show significant overlap between incandescent peak height and color ratio. Each point represents a single particle, and the abbreviations are defined as follows: FA - coal fly ash; VA - volcanic ash; ATD - Arizona test dust; FS. - fullerene soot; Amb. - ambient particles. **(b) Comparison of $FeO_x$ and rBC detection in the SP2.** Relative mass and volume equivalent diameter (VED) for rBC and $FeO_x$ observed by the SP2. Because of the power law relationship for the mass to incandescent signals of $FeO_x$, the largest particles detected by the SP2 are approximately 60x more massive than rBC with the same incandescent signal. $FeO_x$ is significantly denser than rBC, however; the volume equivalent diameter ratio is less than a factor of 3 for particles within the SP2 detection range.

to color ratio relationship for all of the laboratory samples is shown in Figure 2a, along with ambient rBC particles measured in the laboratory.

## 2.4 Ambient measurements

We performed ambient sampling in Boulder, CO from the rooftop inlet of the David Skaggs Research Center to provide samples
5  of typical atmospheric aerosols in an urban environment and natural dust aerosols. The sampling line from the rooftop was an approximately 6 m long, 2.5 cm diameter vertical tube with a small pick-off sampling line of 0.2 cm diameter, which provides a transmission efficiency of ∼100% below 1 micron, and slightly enhances sampling for aerosols >1 micron (super-isokinetic). Observations of ambient aerosols were measured over a period of two days (Oct. 31-Nov. 1, 2018). The sample flow rate of the SP2 during ambient sampling was chosen to be 4 cc/s to match the laboratory data set.

10  ## 2.5 Differentiation of different aerosol types

Classification of aerosols measured by the SP2 rely on differences in both the optical sizes of these aerosols and the aerosols' incandescence and evaporation in the laser beam. Previous work has noted that to first order, $FeO_x$ can be differentiated from



rBC in the SP2 because these aerosols have lower color ratios (Schwarz et al., 2006; Yoshida et al., 2016). Other properties of these aerosols impact their detection in the SP2, including a higher mass to incandescent relationship (Yoshida et al., 2016) and a higher void-free density for $FeO_x$ relative to rBC (5.17 g/cm$^3$ vs. 1.8 g/cm$^3$). This significantly higher mass to incandescent relationship for $FeO_x$ (Yoshida et al., 2016) means that much more massive particles are detected than in the case of rBC with the same incandescent peak amplitude (varying from ~10x to >60x more massive for the largest particles in the SP2 detection range, see Figure 2b). Because the density of $FeO_x$ is nearly 3x higher however, the volume equivalent diameter is only 1.5-3x higher. Literature values for the index of refraction of $FeO_x$ (n=2.30+0.46i, at 1000 nm) also differ significantly from rBC (n=2.49+1.49i, at 1064 nm) (Huffman and Stapp, 1973; Moteki et al., 2010, 2017). Yoshida et al. (2016) noted that the incandescence of hematite occurs deeper in the SP2 laser than magnetite, rBC, Taklamakan Desert dust, and Icelandic dust, likely due to a smaller imaginary part of the index of refraction for hematite than for magnetite.

As has been previously noted, color ratio alone is not sufficient to differentiate rBC and $FeO_x$. In practice, aerosols of the same type show significant statistical variability about their characteristic blackbody temperatures (see Figure 2a). The color ratio measured for single particles is dependent on the particle's mass: color ratios for ambient rBC and fullerene soot at the smallest detectable masses have significantly greater variability due to the lower signal to noise on the red and blue channels. We also found that the width of the distribution of color ratios for rBC and $FeO_x$ as a function of incandescent peak height strongly depended on the alignment of the red PMT detector. The incandescent peak height (mass) and a low color ratio together provide sufficient contrast to differentiate iron oxide containing aerosols from rBC for larger incandescent peak heights (equivalent to ~2 fg rBC) (Yoshida et al., 2016; Liu et al., 2018; Moteki et al., 2017). When detecting mixed populations of $FeO_x$ and rBC, the incomplete contrast between rBC and $FeO_x$ in terms of their incandescent peak heights and color ratio limits detection at smaller sizes. The upper limit of detection for both rBC and $FeO_x$ in the SP2 is due to the gain setting on the detectors; when the incandescent channel becomes saturated, refractory aerosol mass can no longer be quantified.

To improve contrast between rBC and $FeO_x$ at smaller sizes (160-230 nm effective void free diameter for $FeO_x$), core scattering (amount of scattered light measured after volatile coatings have evaporated but before the refractory core has significantly evaporated) can be used as an additional parameter to differentiate rBC and $FeO_x$. Although the real part of the index of refraction for rBC is greater than for $FeO_x$ particles, the incandescent peak height to mass relationship for rBC is also significantly higher (Yoshida et al., 2016). Therefore, $FeO_x$ particles with similar incandescent peak signals compared to rBC are significantly larger particles, and have greater than 4x as much core scattering. However, this additional criteria also does not provide complete contrast between the two aerosol classes at these sizes.

As can be seen in Figure 2, an additional complication arises in differentiating $FeO_x$ from mixed populations of particles including other aerosols with metallic components that incandesce in the SP2, such as natural mineral dust, as the color ratio of ATD, VA, and FA overlap with $FeO_x$ at similar incandescent peak heights. The larger variability in color ratio for ATD, volcanic ash, and coal fly ash in the laboratory samples that were tested is likely due to presence of multiple types of metallic oxides with a greater variety of characteristic blackbody temperatures than $Fe_3O_4$ or $Fe_2O_3$. TEM images have shown that natural mineral dust is composed of numerous grains of different materials, of which iron oxides or other metallic oxides can be one component, typically embedded in quartz or feldspar (e.g. Jeong and Nousiainen (2014)). Yoshida et al. (2016) observed





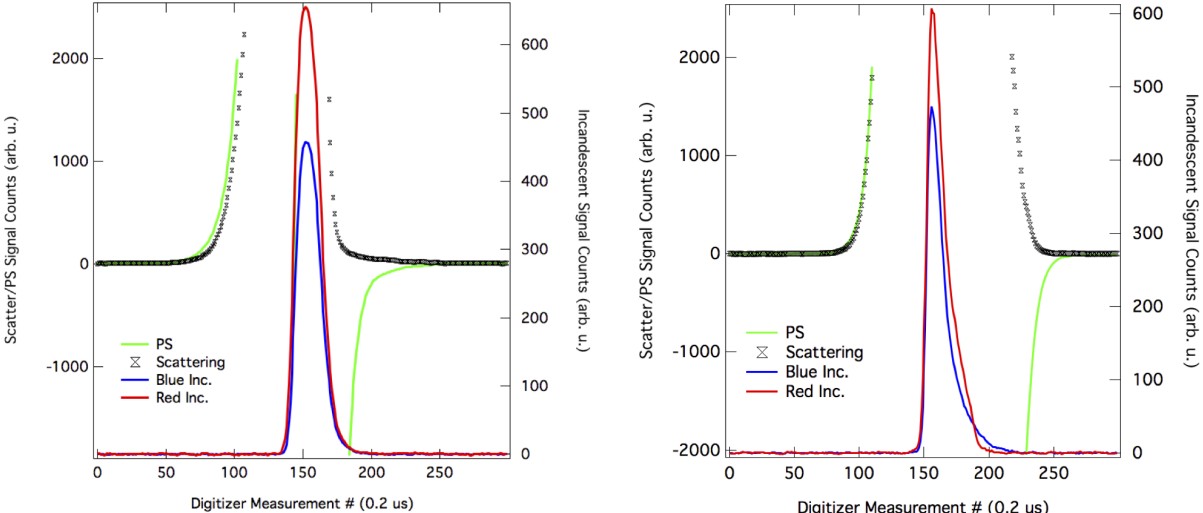

**Figure 3. Comparison of SP2 signals** SP2 traces for two different aerosols with metallic components with similar incandescent peak heights and color ratios. The left signal is from laboratory samples of $Fe_3O_4$ and the right signal is from a mineral dust particle. Black indicates the scattering signal, green is the position-sensitive detector, red is the red PMT signal and blue is the blue PMT signal. For the aerosol particle on the left, the scattering signal disappears as the particle incandesces (around the 200th digitizer measurement point), indicating complete evaporation of the particle. For the particle on the right, scattering is still present after the incandescent signal has returned to the baseline, indicating that non-evaporative portions of the particle still remain after passing through the SP2 laser.

a greater variability in the color ratio for Icelandic dust (similar to the observed color ratios in VA sampled in this study) and speculated that these higher color ratios are due to the presence of strongly light absorbing minerals such as titano-magnetite from volcanic origins. Eyjafjallajökull ash samples were previously found to contain several metallic elements, including similar mass fractions of Fe and Al (Rocha-Lima et al., 2014). SEM-EDS characterization of fly ash demonstrated that typical samples were mainly composed of amorphous alumino-silicate spheres, with a smaller contribution of iron-rich spheres, composed of iron oxides mixed with alumino-silicate (Kutchko and Kim, 2006). On the other hand, anthropogenic iron oxide particles that have been observed in the atmosphere may be coated with organic materials and inorganic materials, but have been found to be predominately metallic (Adachi et al., 2016; Li et al., 2017). (Although coal fly ash is also of anthropogenic origin, we treat these aerosols independently from anthropogenic $FeO_x$, as they have significantly different signals in the SP2, and generally are more similar to ATD.) Core scattering has been shown to be a useful criteria for differentiating anthropogenic iron oxide aerosols from dust-like aerosols (Yoshida et al., 2016; Moteki et al., 2017). However this criteria can only be used on the subset of aerosols that can be optically sized in the typical configuration of the SP2. (Typically $FeO_x$ with effective void-free diameters $< 350$ nm, or $< 230$ nm for the high gain setting used here).

To determine which incandescent signals were likely associated with mineral dust or anthropogenic $FeO_x$, we investigated whether optical scattering in the SP2 can be used to indicate non-evaporative portions of the aerosol (See Figure 3). Given





that natural dust grains typically consist of metallic inclusions surrounded by other materials, even if these portions of the aerosol incandesce in the SP2, the entire particle may not be completely vaporized. Recent measurements using the SP2 to characterize hematite content in dust particles estimated that mineral dust particles detected by the SP2 were generally >500 nm in size (Liu et al., 2018). Post-incandescent scattering, defined as the scattering amplitude of the particle measured after the

incandescent signal has returned to the baseline, indicates portions of the aerosol still remain after the refractory portion of the particle has evaporated (Sedlacek III et al., 2012). Post-incandescent scattering was generally non-zero for both $FeO_x$ and the other aerosols containing metallic components (ATD, VA, FA); in the case of $FeO_x$, the signal was proportional to observed iron oxide mass, which may be related to previous observations that these particles appeared to be melting in the laser beam (Yoshida et al., 2018). Choosing a single threshold value for post-incandescent scattering only differentiated $FeO_x$ from other

aerosols containing metallic components for ~80% of the particles, however.

## 3   Supervised learning methods applied to aerosol classification with the single particle soot photometer

The inability to unambiguously classify $FeO_x$ from anthropogenic sources from either natural mineral dust or rBC using a small number of features (the incandescent peak height, the color ratio, core scattering, and post-incandescent scattering) derived from the single particle signals suggests that new analysis approaches should be explored to fully exploit this additional aspect

of the SP2. The SP2's limitations in classifying different aerosols could be overcome in some cases by changes to the detection scheme, e.g. detectors with greater dynamic range for optical sizing; however, other limitations are fundamentally linked to the L-II method, as there is an overlap between the features associated with different particle types when only those 4 features are considered. Previous data sets (e.g. aircraft data sets) are also limited to the detection scheme described in Section 2, so it would be beneficial to formulate a method to use this data.

From a mathematical perspective, the problem of classifying aerosols can be described as the search for a mapping function $f$ that maps a feature vector $\mathbf{x}_i$ associated with each aerosol to its correct class label $y_i$. Here we define a feature as an attribute (for example, the incandescent peak height) associated with a single particle $i$, which can be expressed as an n-dimensional feature vector $\mathbf{x}_i \in \mathbb{R}^n$. We would like to find a separable subspace within the n-dimensional feature space that can be used to differentiate aerosols by class. In other words, decision boundaries can be found that allow us to separate the different aerosols

with minimal misclassifications. Decision boundaries are hyperplanes of dimension $n-1$ that subdivide this feature space such that the different classes (rBC, $FeO_x$, etc.) reside in distinct subspaces.

    Supervised learning algorithms are a class of machine learning algorithms that map input variables ($\mathbf{X}$) to a predicted output variable ($\hat{\mathbf{Y}}$), after first training the algorithm using a set of input variables ($\mathbf{X}'$) with known output ($\mathbf{Y}'$). (Here we adopt notation to use $\hat{y}_i$ to differentiate the predicted label from the actual label $y_i$. For $n$ training examples and $m$ features, the

input vectors generalize to matrices.) Following the notation in Mohri et al. (2012), the problem that the learning algorithm attempts to solve is finding a hypothesis h, where h $\in H$ (a subset of functions explored by the learning algorithm), to map $\mathbf{x}_i$ to a predicted class label $\hat{y}_i$, such that the loss function $L(\hat{y}_i, y_i)$ is small. This loss function $L(\hat{y}_i, y_i)$ gives the cost of predicting $\hat{y}_i$ rather than $y_i$. We would like to avoid hypotheses that either "underfit" or "overfit" the data. Underfitting refers





to a hypothesis that does not perform well even on the initial training data set, as it does not capture the trend of the data. Overfitting occurs when a hypothesis fits the training data well, but cannot generalize to new cases, because it has too closely constrained the model to the specific data set. Over-fitting can be addressed by increasing the number of training examples (to provide greater instances of within class variance), while underfitting generally implies that the chosen features do not allow

for enough degrees of freedom, and a more complex model is required. Since the small set of features described in Section 2 do not provide enough information (e.g. they "underfit" the data), we would like to expand the number of features that provide some information about what type of aerosol was observed. However, adding additional features creates a much larger space over which to determine appropriate decision boundaries to differentiate aerosols. Moreover, the large variation within classes (between aerosols of the same type with different masses or internal mixing states) makes this problem even more challenging.

This kind of problem is tractable using supervised machine learning, however; and these algorithms can be readily applied using existing software libraries (e.g. Python's scikit-learn and TensorFlow libraries).

There are a variety of machine learning algorithms that can be used for classification; the choice of an algorithm depends on consideration of both the particular data set and the intended application, as there's no clear cut superiority of performance between different algorithms. Here we focus on the the application of a random forest algorithm to the SP2 observations, as its

performance on classifying aerosols measured with the SP2 was found to be superior to other methods (e.g. neural networks, dimensionality reduction, etc.) that were tested using the approach outlined in this study. A random forest consists of an ensemble of decision trees, and is described in greater detail in Section 3.3. This approach allows us to extend the number of features considered for an individual particle to improve classification performance. We compare two different approaches for applying this random forest algorithm to the laboratory data. In one case, we use 6 distinct classes (rBC, ATD, FA, VA, $Fe_2O_3$,

and $Fe_3O_4$), where rBC includes both bare and coated fullerene soot in the training data set. In the second case, we use only 3 distinct classes: rBC (again, including coated and uncoated fullerene soot as training data), dust-like aerosols (ATD, VA, and FA), and $FeO_x$ ($Fe_2O_3$, and $Fe_3O_4$).

The application of (any) supervised machine learning algorithm requires the implementation of several steps: first, data is collected, and in the case of supervised learning algorithms, labeled and randomly separated into independent training, cross-

validation, and test sets. The randomly selected training data set is needed to train the model, the cross-validation set is used to determine the optimal set of hyperparameters for the algorithm, and an independent test set provides information about how the trained algorithm generalizes to new cases. Before the application of the algorithm to the data set, however, the data needs to be preprocessed, which entails repairing or removing missing values and transforming variables by normalizing and scaling them. In the case of "classic" machine learning, features are extracted from the data set (more advanced techniques such as

representation learning/deep learning operate directly on raw data to extract features, but for computational simplicity we do not explore this approach here (Goodfellow et al., 2016)). The third step is training the algorithm on the training data set and optimizing its performance using the cross-validation set. The fourth step is evaluating the performance of the algorithm on the test set. Finally, the trained algorithm can be applied to new data sets. The computation steps required to train and optimize the algorithm and then apply it to new data is referred to as a machine learning pipeline. We describe the first three steps in this

section, and discuss its performance on the test data set and on atmospheric measurements in Section 4.





## 3.1 Feature engineering from single particle signals

The typical analysis method for the NOAA SP2 reduces 80 $\mu$s time series signals to a vector of features that can be used to
determine the mass, optical size, coating state, and coating thickness of rBC given appropriate calibrations for the detectors as
was described in Section 2. An algorithm is applied to filter out signals that may be contaminated, e.g. by multiple particles
measured during the same acquisition window or other non-ideal sampling conditions.

To leverage existing SP2 feature engineering and data analysis, we consider a number of features derived from the single
particle signals, including features previously demonstrated to provide useful information about the particle's physio-chemical
properties. The features explored in this analysis are shown in Table 2 and for an example rBC particle in Figure 4 and can
be roughly divided into three categories: those associated with the incandescent channels ($x_0$-$x_1$), those associated with the
scattering channels ($x_2$-$x_7$), and those derived from the timing of different signals in the beam ($x_8$-$x_{16}$). Those associated with
the scattering channel are related to the optical size of the aerosol as it traverses the laser (core scattering, total maximum
scattering, position sensitive wideness, post-incandescent scattering, and the optical size at a fixed point along the evaporating
edge). Those associated with the incandescent channel relate to the mass and thermal properties of the aerosol (the blue
peak amplitude and the color ratio). Those associated with timing in the beam (e.g. min. scattering before incandescence,
incandescent start position, evaporation point, incandescent used length, and incandescent total length) are related to both the
size and physio-chemical properties of the particle (e.g. whether it is initially coated with any volatile materials, how strongly
absorbing the aerosol is, or how long it takes to evaporate in the laser); several of these features also depend on the specific
laser settings. It is necessary to point out that the features derived from the single particle signals in some cases will not provide
values directly interpretable as a measure of the physical properties of the particles as given in Table 2 (i.e. for larger particles,
the detectors may be saturated). If the systematic bias with the feature or measurement artifacts are repeatable however, they
still provide useful information to the algorithm, as these methods rely on statistical relationships between the data sets rather
than an underlying physical model. To quantify detector saturation for larger particles, we have included features that indicate
whether the scattering signal is high before the start of incandesce ($x_7$) and how long the signal is saturated ($x_{10}$).

These features were chosen because they generally showed some separability for different aerosol types for the laboratory
samples, although no single feature or pair of features was sufficient to entirely separate different aerosol classes. In applying
the machine learning algorithm for the two different cases, we initially use all features, but also explore whether a reduced set
of these features can provide similar classification performance.

## 3.2 Data preprocessing

Many machine learning algorithms work best if features are preprocessed so that they are normally distributed and have a mean
of 0; however, one advantage of decision trees (and by extension, random forests) is that they are fairly robust to feature scaling
and normalization. We perform several preprocessing steps to prepare the data for use in the algorithm; first, we remove data
associated with particles that do not at minimum have values for both the incandescent peak height and color ratio. For features



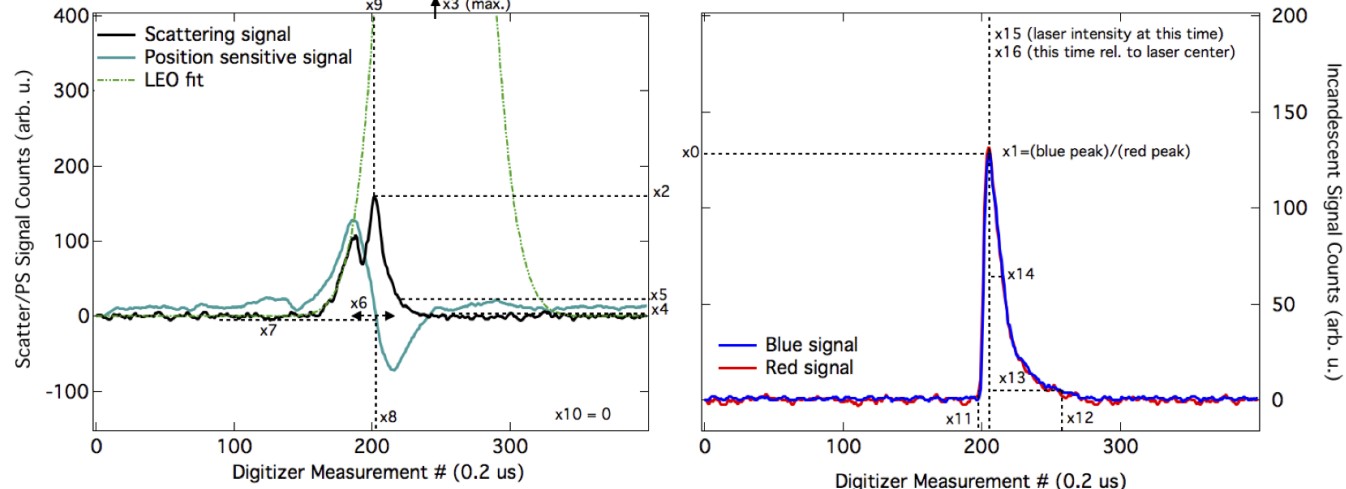

**Figure 4. SP2 signal for a single particle showing all features used in the algorithm** SP2 traces for a coated rBC particle (4 fg, 20 nm thick coating assuming $n_{core}$=2.26+i*1.26 and $n_{coat}$=1.45) showing the features used in the machine learning algorithm. The left figure shows the scattering and position sensitive detectors and the right figure shows the blue and red incandescent channels for the same particle. Physical interpretation and descriptions of these features are further detailed in Table 2. Since the scattering signal is not saturated at any point for this particular particle, $x_{10} = 0$. $x_3$ is proportional to the maximum value of the scattering signal derived from the leading-edge only (LEO) fit (Gao et al., 2007).

that are expected to be log-normally distributed (the blue signal amplitude, which is proportional to the particle mass, and the features associated with different optical sizes), we also take the natural logarithm in order to have normally distributed values.

The other important step in preparing data for supervised learning is figuring out what to do with particles associated with incomplete information. In certain cases (such as when the scattering detector is saturated for larger particles), the unknown
number likely represents known information (e.g. that the particle is too large to optically size). For these four features (core scattering, scattering peak amplitude, evaporation scattering size, and the laser intensity at the peak of incandescence) we assume that missing values are larger than any of the values that were recorded; therefore these missing values were imputed with 110% of the highest expected value for each of those features. For the other 13 features, we used dummy values outside the typical detection range to effectively exclude these features for that particular sample (alternatively, we tested using typical
mean values for these features, which led to similar classification performance). Preprocessing could potentially be improved by implementing a learning algorithm to optimize the imputation of missing values; however, because we did not observe a significant change in performance dependent on the particular method for preprocessing, we did not explore this option.

### 3.3 Random Forest

After feature selection and data preparation, we apply an algorithm that can construct a model from the data based on the
15 samples in the training data set. In this case we use a random forest algorithm, which consists of an ensemble of decision trees.



**Table 2. Description of features from processed SP2 single particle signals** For features that correspond to times in the SP2 detection window, the time is referenced to the position-sensitive detector cross-over point, which is a fixed point in space (independent of particle size).

| Symbol | Feature | Description/Physical interpretation |
|---|---|---|
| $x_0$ | Blue peak amplitude | Function of aerosol incandescent mass |
| $x_1$ | Color ratio | Function of aerosol blackbody temperature |
| $x_2$ | Core scattering | Function of optical size of core after volatile coatings have evaporated |
| $x_3$ | Total scattering max. | Function of optical size of the aerosol including coating (estimated from LEO fitting) |
| $x_4$ | Post incandescent scattering | Function of optical size of aerosol after refractory portion has evaporated |
| $x_5$ | Evaporation scattering size | Function of scattering ampl. along evaporating edge when inc. signal ∼1 fg rBC |
| $x_6$ | Position sensitive wideness | Time difference between max. and min. of PS signal amplitude |
| $x_7$ | Min. scattering before incandescence | Minimum value of scattering signal before start of incandescence |
| $x_8$ | Position sensitive trigger position | Time of PS detector cross-over point (change from positive to negative) |
| $x_9$ | Scatter peak location | Time of maximum scatter signal |
| $x_{10}$ | Saturation width | Total time scattering signal is saturated, relative to Gaussian laser width |
| $x_{11}$ | Incandescent start position | Time inc. signal is first greater than min. threshold (relative to PS cross-over point) |
| $x_{12}$ | Evaporation point | Time at which aerosol has completely evaporated (relative to PS cross-over point) |
| $x_{13}$ | Incandescent total length | Time from incandescence peak location until signal has decayed to baseline |
| $x_{14}$ | Incandescent used length | Full width half maximum of incandescent signal |
| $x_{15}$ | Light on laser intensity | Laser intensity calculated at blue maximum peak location |
| $x_{16}$ | Width fraction from center | Inc. peak location relative to center of laser, scaled by Gaussian laser width |

Each decision tree is a classifier that works by sequentially subdividing the training data set based on learned threshold values of the features at each node (See Figure 5, top panel). At each node, the threshold values are determined by minimizing the impurity $F$ over the classes associated with the subset of samples in the two resulting "children". Typical measurements of node impurity $F$ are the information entropy or the Gini index (Mohri et al., 2012). Information entropy provides a metric for quantifying how much information is in an event; for example, decisions that split the training samples such that a single class is represented in each child have lower entropy than splits resulting in multiple classes, since there is greater information gain. The Gini index measures the likelihood that a randomly chosen sample would be mislabeled given the values of the labels in the subset associated with each child. After a sufficient purity according to one of these metrics has been reached, the algorithm is stopped. The class associated with the majority of the training samples after the terminal split along any particular branch of the tree is associated with that "leaf", and new samples that satisfy the same criteria are predicted to have that class.

Decision trees are fairly robust even for cases of features that are not normally-distributed. They have the advantage of having few tunable parameters, meaning that their out-of-the-box implementation is simpler than many other machine learning algorithms. They can also directly handle multi-class classification problems such as the one we consider here. They are



non-parametric machine learning algorithms, i.e. no *a priori* assumptions are made about the function to be learned, and the complexity of the model is a function of the training data set size (Goodfellow et al., 2016). These types of algorithms do typically require more training data and longer training times than parametric models, but can generally result in more powerful models. However, a common problem for decision trees is overfitting, meaning that their generalization to new examples can be poor; even small changes in the training data set can lead to different outcomes.

Random forests improve upon the performance of single decision trees by growing an ensemble of decision trees. This algorithm is more robust to overfitting than a single decision tree, as they rely on bagging ("bootstrap aggregrating"): a random selection of the training samples are chosen to grow each decision tree, with replacement from the original training set. Each tree also uses a random selection of a subset of the features to define the split at each node (Breiman, 2001). Because of this randomness, the generalization error converges for a large number of trees. The predictions of the random forest are based upon the ensemble vote from all the trees in the forest; that is for each sample, the trained algorithm outputs a conditional probability distribution $f(y_i) = p(y|\mathbf{x}_i, \theta)$ over the classes, with the highest probability corresponding to the most likely class of the particle $\hat{y}_i$, given the values of the feature vector $\mathbf{x}_i$ and the optimized parameters of the algorithm $\theta$. In the implementation of the algorithm used here, the conditional probability for a new sample to belong to any particular class is determined by averaging the probability predictions from all the trees in the forest (Pedregosa et al., 2011). This algorithm has previously been used to classify single particle mass spectra (Christopoulos et al., 2018). A schematic for applying the random forest to a single particle signal from the SP2 is shown in the bottom panel of Figure 5.

### 3.4 Computational resources

For this analysis, we used Python 3.6.6 with the sci-kit learn package version 0.20.0. To train the algorithm and optimize its hyperparameters, we used a remote Linux server with 24 cores (2 Intel Xeon CPU E5-2695 v2 2.40 GHz processors with 12 cores each and two threads enabled per core), which provided ∼125 GB of RAM memory.

The computation time for training a random forest is directly related to the number of trees in the forest and the amount of training data used. In general, a binary decision tree has a time complexity of $\mathcal{O}(mn\log n)$ for $n$ training samples and $m$ features (Pedregosa et al., 2011). One disadvantage of this method is that each decision tree needs to use all of the training data (or rather, the subset of the training data used to train that tree) at once in order to grow the tree, requiring a significant amount of memory for large training data sets. Distributed or parallel methods can be used to improve computation efficiency for random forests, as the trees can be grown simultaneously on different processors. Depending on the complexity of the trained random forest, the decision path of the trained algorithm can also require significant memory to store. With the large training data set that we consider here, this method is computationally expensive, so although this approach serves as a proof of concept, it would be advantageous to explore other methods for improved computational efficiency.

### 3.5 Tuning hyper-parameters for improved performance

In order to optimize the performance of supervised machine learning algorithms, a cross-validation data set is typically used to find the optimal set of hyper-parameters for the algorithm. Hyper-parameters are separate from the parameters optimized by





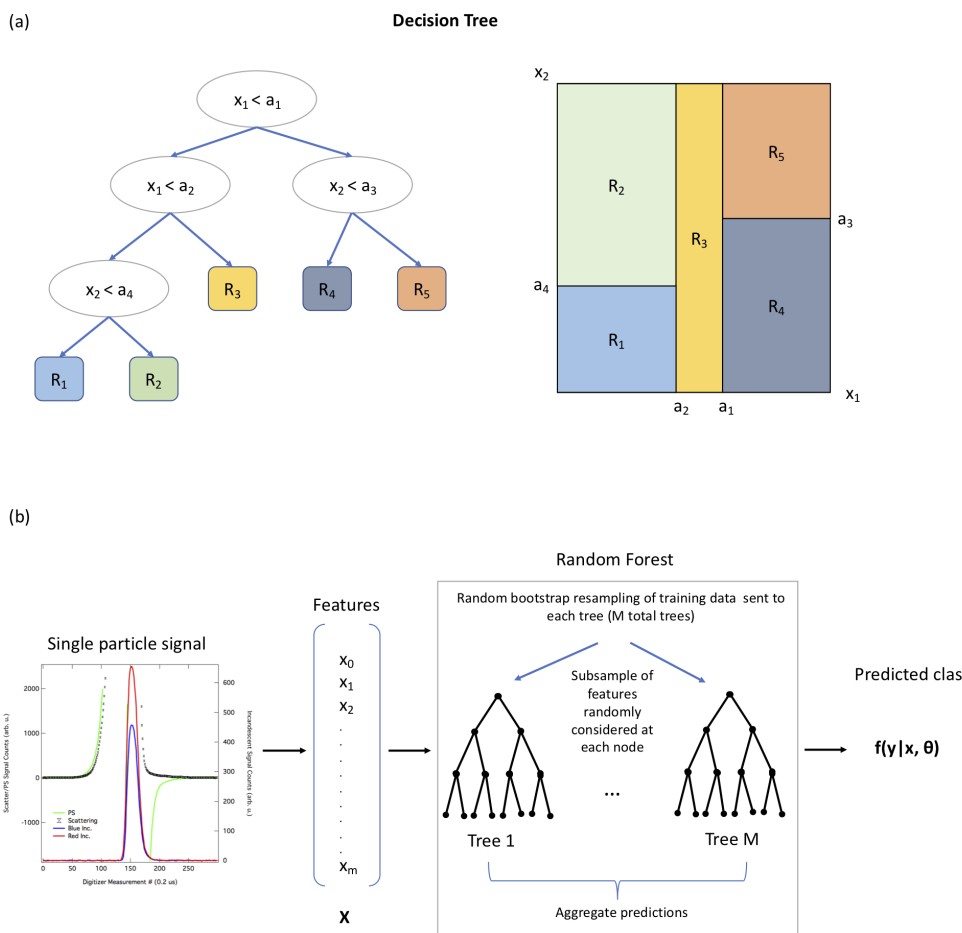

**Figure 5. Schematic of a decision tree (top), and a random forest as applied to the SP2 signals (bottom).** *(a)* A simplified example of a decision tree for a case with only 2 features, demonstrating how choosing threshold values of the features at each node (left) subdivides the feature space into different regions (right). The true case corresponds to the left "child" in the tree. This figure has been adapted from Mohri et al. (2012). *(b)* The schematic for applying supervised learning to predict the aerosol class for a particle measured by the SP2 is shown. First, single particle signals are processed and reduced to a feature vector $\mathbf{x}_i$, which is then given to the trained random forest. Each decision tree within the forest makes a series of decisions based on the values of the features (which have been learned from a random subset of the training data, by considering a random subsets of features at each split) to predict the most likely class of the particle based on the observed features. The aggregated prediction from the ensemble of $M$ decision trees predicts a probability distribution $f(y) = p(y|\mathbf{x}_i, \theta)$ over each class $y$.





the learning algorithm, but affect its generalization performance. For a random forest, the hyper-parameters consist of e.g. the number of trees in the forest (number of estimators), the maximum depth of each branch, and the maximum number of features (the subset of features) randomly considered at each node. Several of these hyper-parameters affect the growth of individual trees, serving as alternative stopping criteria, by e.g. setting a lower limit on the number of training samples required for each

split, or setting an upper limit on the depth of the tree.

Random forests can use out-of-the-bag (o.o.b.) error estimates for determining hyper-parameters (Breiman, 2001). Since each tree relies on only a sub-sample of the training data set, the performance of that tree on the subset of the training data not used in growing the tree can be used to optimize the hyper-parameters. We used o.o.b error estimates to initially tune over a number of different hyper-parameters to see which most impacted the classification performance. We found that the entropy

criteria always performed better than using the Gini index as the metric for optimizing splitting at each node. Increasing the number of estimators also always improved the classification accuracy, although the effect reached an asymptote after ∼100-120 trees. The maximum depth of each tree showed decreasing o.o.b error for greater depths, up to ∼40 splits in the 3 class case, and ∼ 30 splits in the 6 class case, with very little change for greater depths. Having fewer samples required to allow for an internal split and fewer samples allowed per leaf also tended to decrease error. The maximum number of features randomly

considered at each split had the strongest difference between the 6 class and 3 class cases, with the 3 class case performing better with a smaller subset of features than the 6 class case.

Since each hyper-parameter does not independently impact the performance of the algorithm, we then used a grid-search with 3-fold cross validation over 54 different combinations of the hyper-parameters to find the best set for each case (Mohri et al., 2012). K-fold cross-validation uses a subset of the training data (after the data has been randomly shuffled, and with

equal relative sub-selections of each aerosol class) as a cross-validation data set (typically K ∈ [3, 10]), while the algorithm is trained on the rest of the data. The procedure is repeated K times, using a different subset of the data each time as the cross-validation set in order to get K total values for the classification accuracy for each set of hyper-parameters. For the 6 class case, the optimal hyper-parameters were a max. depth of 50, 12 features considered at each split, a minimum of 1 sample per leaf, and a minimum of 2 samples per split, with 120 estimators, and using entropy as a criterion. For the 3 class case, the

optimal hyper-parameters were a max. depth of 50, 10 features considered at each split, a minimum of 1 sample per leaf, and a minimum of 4 samples per split, with 120 estimators, and using entropy as a criterion.

## 4   Performance on laboratory samples and atmospheric observations

The performance of the trained and optimized random forest was tested on an independent test set of laboratory samples. Testing the trained algorithm on an independent set of labeled data allows us to determine its generalization performance (the

skill of the learning algorithm to classify new particles) before applying it to the (unlabeled) atmospheric data set. We used stratified K-folding (which randomly divides data with equal sub-selections of each aerosol class) to divide the laboratory data set into 3rds, with 2/3rds used for the training/cross-validation steps, and with 1/3rd kept aside as a test data set.





## 4.1 Improving the pipeline

After optimizing the algorithm, we investigate whether the machine learning pipeline can be improved. The importance of different features (given in Table 3) is determined by removing any single feature and running the algorithm to see how the classification accuracy on the test set changes. The importance of each feature determined using this method only gives

information over the entire population of aerosols in each class in the test set, however. For both the 6 class and 3 class cases, the color ratio is the most important feature, with the post-incandescent scattering being the next most important feature. As both of these features were previously identified as physically meaningful for separating different aerosol types, this is not surprising.

To avoid overtraining our algorithm, we would ideally like to use the smallest set of features possible. We remove the least

important features (retaining the 11 most important features for the 6-class case and the 9 most important for the 3-class case), and retrain and optimize the algorithm in each case. We chose these reductions since there was a clear break in the relative importance of different features at these points for each case.

Removing additional features (e.g. using only the top 5 most important features for the 3 class case) still performed well for $FeO_x$ and rBC, but led to significantly worse classification accuracy for the dust-like aerosols, suggesting the additional

features provided enough information to reduce the misclassification of dust-like aerosols from rBC by a factor of 2. Another advantage of reducing features is that the algorithm can be trained faster (in the case of the 6 class case, the training time was reduced by approximately 1/3rd when using 11 rather than 17 features for each sample).

## 4.2 Confusion matrices

To quantify the performance of the algorithm for each of the cases, we visualize the true positive and false positive rates for

each class using a confusion matrix (Figure 6). Confusion matrices are useful ways to visualize how well a classifier performs. For each class, they give the number of particles of that class that are predicted to belong to that class (the true positives, along the diagonal) vs. all of the misidentifications of the particles as other classes (the false positives, the off-diagonal elements of the matrix). Since our test data set does not have the same number of particles for each class, we normalize along each horizontal row to give the fractional portion predicted for each of the class labels.

For both the 6 class and 3 class cases, we find the worst performance for the aerosols containing metallic inclusions (ATD, VA, FA - the "dust-like" aerosols). These aerosols are most likely to be misidentified as either rBC or as one another. One reason for this may be the imperfections of the laboratory data sets. Previous work has noted that there is a small fraction of rBC present in laboratory samples of ATD (S. Kaspari, private communication), which likely contributes to the high rate of errors between ATD and rBC. Additionally, because of the low incandescent rate of these aerosols, we had acquired the fewest

number of training examples for these data sets. In general, these aerosols are much less likely to be misidentified as $FeO_x$ (1-3% false positives) by the trained algorithm than as rBC, however.

We also find that rBC is unlikely to be misidentified as any of the other aerosol types, including $FeO_x$. For the 6 class case, $Fe_3O_4$ was more likely to be correctly identified than $Fe_2O_3$; this could be because some portion of the $Fe_2O_3$ is more similar



**Table 3. Importance of different features.** The relative importance of the different features for the optimal random forest for the 6-class and 3-class cases. All denotes that all 17 features were used in the algorithm, and reduced indicates that only a subset of features were included as input to the learning algorithm (11 features for the 6 class case and 9 features for the 3 class case). The top 5 most important features in each category are bolded.

| Feature | All (6 classes) | Reduced (6 classes) | All (3 classes) | Reduced (3 classes) |
|---|---|---|---|---|
| Blue peak amplitude | **0.064** | **0.075** | **0.049** | **0.057** |
| Color ratio | **0.391** | **0.358** | **0.455** | **0.587** |
| Core scattering | 0.032 | 0.044 | **0.051** | 0.019 |
| Total scattering max. | 0.021 | • | 0.014 | • |
| Post incandescent scattering | **0.148** | **0.190** | **0.183** | **0.136** |
| Evaporation scattering size | **0.042** | **0.053** | **0.043** | 0.037 |
| Position sensitive wideness | 0.018 | • | 0.011 | • |
| Min. scattering before incandescence | 0.023 | • | 0.020 | • |
| Position sensitive trigger position | 0.024 | • | 0.017 | • |
| Scatter peak location | 0.026 | 0.045 | 0.019 | • |
| Saturation width | 0.028 | 0.046 | 0.015 | • |
| Incandescent start position | 0.039 | **0.054** | 0.029 | **0.057** |
| Evaporation point | 0.035 | 0.052 | 0.025 | **0.054** |
| Incandescent total length | 0.030 | 0.036 | 0.023 | 0.028 |
| Incandescent used length | **0.044** | 0.047 | 0.026 | 0.025 |
| Light on laser intensity | 0.018 | • | 0.011 | • |
| Width fraction from center | 0.016 | • | 0.011 | • |

to $Fe_3O_4$, or perhaps because of the imbalance of training examples for the two aerosols. The confusion matrices make it clear that $Fe_3O_4$ and $Fe_2O_3$ are more likely to be misclassifed as one another than as any of the other aerosol types.

### 4.2.1 Base case

To provide a basis for the comparison of the different supervised learning algorithms, we also consider a classification scheme which uses only a few features and linear decision boundaries to classify incandescent aerosols. We designate this case the "base case", and use only the color ratio and incandescent peak height to differentiate rBC from $FeO_x$, and the additional criteria of the core scattering to differentiate anthropogenic $FeO_x$ from mineral dust with metallic inclusions. This is based on the method that has previously been used in e.g. Moteki et al. (2017); Ohata et al. (2018); Yoshida et al. (2018).

We visualize the simple pathway through feature space using this method in the top panel of Figure 7. The threshold values for the metallic mode (including both dust-like and $FeO_x$ samples) for the incandescent peak height was chosen to be 2 fg rBC equivalent-mass and the color ratio margin was chosen to be 0.785 from inspection of the modes in the data. We designate the



**Figure 6. Confusion matrices for the 6-class and 3-class cases** A visualization of the confusion matrices using all features (left matrices) and the reduced feature space (right matrices) is shown for the laboratory test data. The y-axis in each figure indicates the true particle type and the x-axis indicates the particle type predicted by the trained random forest on the test data set. The fraction of aerosols identified as a particular class relative to the total number of aerosols is given, with the actual number of particles shown in parentheses.

region used for rBC as Region 1 (regions are shown in the top right panel of Figure 7). As discussed in Section 2.5, we can only optically-size aerosols in the metallic mode for incandescent peak height between ∼2-4 fg rBC equivalent mass; this portion of the metallic mode is designated as Region 2. Since we cannot optically size these aerosols outside of this range (Region 3), we need an additional assumption in order to estimate classification performance. Previous work has only designated $FeO_x$ as

5    anthropogenic or dust-like for aerosols in Region 2, and used this (and off-line techniques) to provide context for interpreting





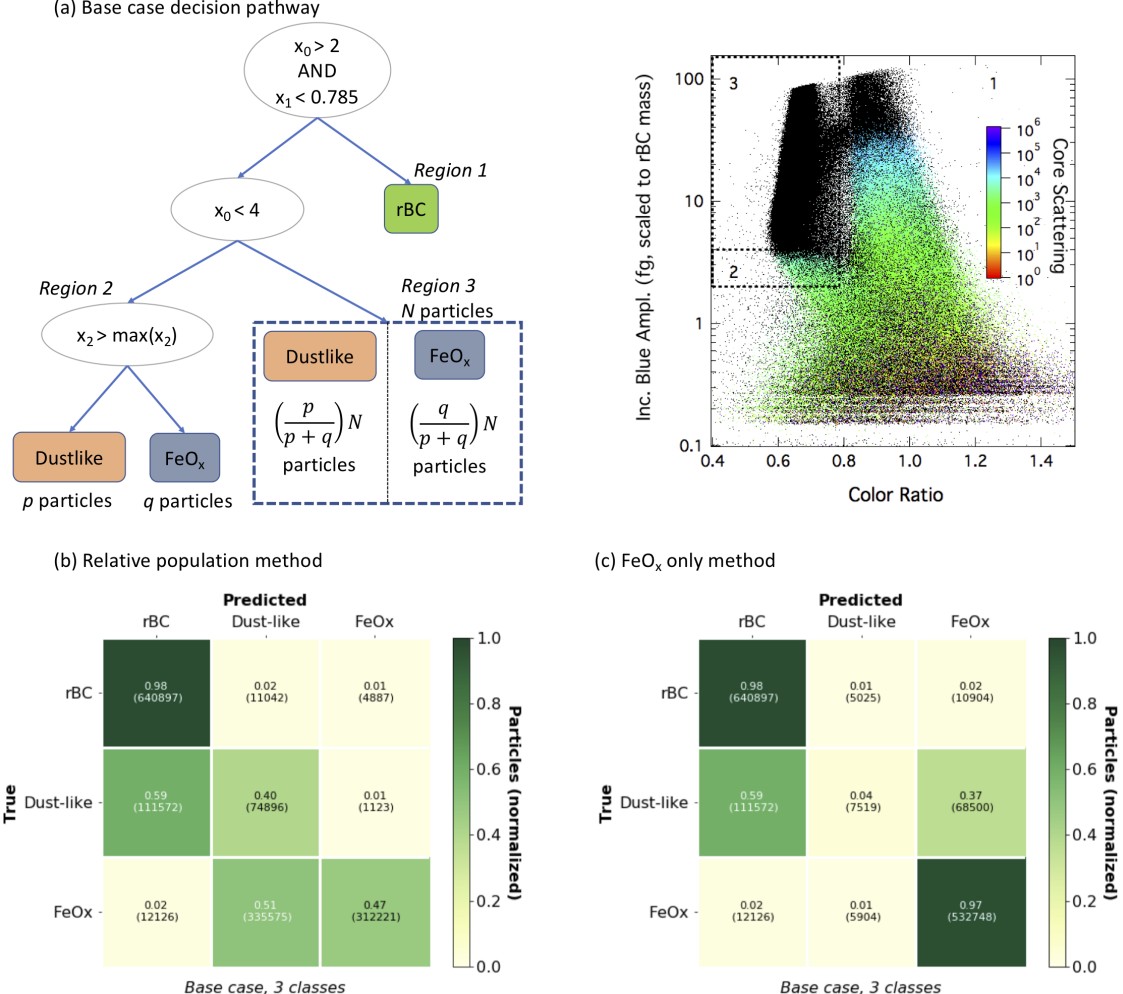

**Figure 7. Base case classification scheme and confusion matrices.** *Top:* We visualize the scheme for classifying aerosols using a simplistic pathway through feature space. The features are the same as in Table 2, and $x_2 > \max(x_2)$ indicates the detector is saturated for the core scattering measurement. The true case at each node corresponds to the left "child". We need to make an additional assumption for the particles in the metallic mode that cannot be optically sized (the right-most split, corresponding to Region 3). Saturated values for core scattering are indicated by black dots in the right figure, which shows the incandescent peak height vs color ratio for the laboratory samples. *Bottom, left:* Confusion matrix for the base case assuming the same ratio of anthropogenic $FeO_x$:dust-like in Region 3 as measured in Region 2. Since the particles in Region 3 are differentiated only on a population basis, to visualize this case as a confusion matrix, we have made the additional assumption that particles are correctly classified up to the total number of aerosols of that class (any additional aerosols identified as that class are assumed to be false positives). *Bottom, right:* Confusion matrix for the base case identifying all aerosols in Region 3 as anthropogenic $FeO_x$, to provide an upper limit on anthropogenic $FeO_x$.





atmospheric observations of $FeO_x$ in East Asia as dominated by anthropogenic emissions (Moteki et al., 2017; Ohata et al., 2018; Yoshida et al., 2018).

To demonstrate how well the base case provides information for the entire population of aerosols measured by the SP2, we consider two possible assumptions for Region 3 using the laboratory samples; we emphasize that these results are specific

to the distributions and relative numbers of aerosols in the laboratory samples and should be taken as illustrative only. One reasonable assumption would be to assume that the entire population of aerosols in the metallic mode would have a similar ratio of anthropogenic to dust-like $FeO_x$ as the aerosols in Region 2 (relative population assumption). However, this assumption clearly relies on the number distributions of dust-like and anthropogenic $FeO_x$ aerosols being similar over the entire range that the SP2 can measure (which would be unknown in ambient populations). Another assumption would be to take all the particles

in Region 3 as an upper limit on anthropogenic $FeO_x$ ($FeO_x$ only assumption).

We visualize the classification performance for the base case under these two assumptions as confusion matrices, as shown in the bottom panel of Figure 7. Since it's not possible to identify the 6 individual aerosol types using this method, we show only the 3 class case for this scheme. The relative population assumption leads to a significant underestimation of anthropogenic $FeO_x$, as can be see in Figure 7b. (Since the relative population method does not classify individual aerosol particles, to

visualize this scheme as a confusion matrix, we assume aerosols are true positives up to the number of total aerosols of that class in our data set, and otherwise are false positives.)

For the $FeO_x$ only method (Figure 7c), the true positive rate for rBC and $FeO_x$ is still quite good (98% and 97% respectively), but the false positives are significantly more problematic than for the supervised machine learning classification schemes. Nearly 2% of the rBC is misidentified as $FeO_x$, which could be problematic given that only a small fraction ($\sim$1/250) of

ambient aerosols in urban areas that incandesce in the SP2 are expected to be $FeO_x$. The misclassification of the dust-like aerosols is even more problematic, as nearly 37% of these aerosols would be identified as $FeO_x$ for this particular data set. Given that in ambient populations we would expect to have no prior knowledge about the relative proportion of anthropogenic to dust-like aerosols, this large misclassification rate could significantly bias the interpretation of the $FeO_x$ mass mixing ratio in certain cases.

**4.3  Precision and recall**

The confusion matrices only provide information about how well the trained algorithm performs on each class in general. However, from a measurement perspective, we are interested in whether there might be significant bias for different classes in certain cases, such as aerosols with smaller or larger incandescent masses. To investigate how well the algorithm performs as a function of the incandescent peak height, we define two metrics, the precision and recall. The precision $P_i$ for a particular

class $i$ is defined as

$$P_i = \frac{\text{\# true positives}}{(\text{\# true positives}) + (\text{\# false positives})} \qquad (1)$$

and recall $R_i$ for class $i$ is defined as

$$R_i = \frac{\text{\# true positives}}{(\text{\# of true positives}) + (\text{\# false negatives})}. \qquad (2)$$





True positives are defined as aerosols of class $i$ that are predicted to belong to class $i$. False positives are aerosols of other classes ($j \neq i$) that are predicted to belong to class $i$. False negatives are aerosols of class $i$ that are predicted to belong to a different class $k$ ($k \neq i$). Precision provides information about how accurately the algorithm identifies aerosols of a particular class, whereas recall provides information about how many of the relevant particles are actually identified. For FeO$_x$, we are

most interested in maximizing the precision (as opposed to the recall), i.e. we would like to avoid falsely identifying other types of particles as FeO$_x$. Lower recall would translate to an underestimation of the mass mixing ratio of FeO$_x$, whereas having lower precision could introduce a significant systematic bias due to other aerosols being misidentified as FeO$_x$.

Figure 8 shows a side-by-side comparison of the performance of the 6 class (reduced features) case and 3 class (reduced features) case on the laboratory test set. The true and predicted labels for the aerosols, shown as a function of the incandescent

peak height vs. color ratio is shown. Comparing the true labeled data set with the predicted labels indicates that the portion of the dust-like aerosols that overlap with the rBC population is the one that is most likely to be misidentified. As stated previously, this may be due to a fraction of rBC present in the laboratory samples of ATD. The precision and recall for each of the aerosol classes is also shown, binned as a function of the incandescent peak height. In general, the performance of the classifiers are better for FeO$_x$ at larger masses than at smaller masses; this is likely impacted by the size distribution of the aerosols in our

training data set.

### 4.4    Performance on atmospheric observations

We finally apply the machine learning pipeline for the 3 class, reduced feature case to the atmospheric data sets acquired in Boulder, CO (Figure 9). Generally we observe the three modes that we expect for rBC, dust-like aerosols, and FeO$_x$ based upon the predictions of the random forest algorithm. These observations indicate that the laboratory samples in general have

similar responses in the SP2 as the aerosols that we are observing in the urban environment. This suggests that the algorithm identifies iron oxides sourced from anthropogenic emissions with the laboratory samples of pure iron oxides. This application demonstrates that a significant feature of FeO$_x$ is present in the atmosphere in Boulder, as has previously been observed in urban areas in East Asia (Yoshida et al., 2016; Moteki et al., 2017; Ohata et al., 2018).

The algorithm also identifies a significant fraction of dust-like aerosols, both at cooler color temperature ratios, and mixed

into the population of aerosols in the rBC mode. At the larger masses for the rBC color ratio mode, the algorithm does appear to be misidentifying a fraction of the rBC with cooler color temperature ratios as dust-like aerosols, perhaps due to the sensitivity of the specific values of the color ratio to the alignment of the PMT detectors or differences between the SP2 response to ambient rBC vs. fullerene soot. This result suggests that the algorithm has been overtrained to the specific threshold values for the color ratio in this case.

Previous studies have indicated that a fraction of rBC present in the atmosphere may be attached to other aerosols such as natural dust ("attached-type rBC"), rather than present in a core-shell structure ("coated rBC") (Sedlacek III et al., 2012; Dahlkötter et al., 2014; Moteki et al., 2014). In the ambient measurements in Boulder, we found $\sim$7% of the aerosols in the rBC mode were predicted to be dust-like (excluding color ratios below 0.85 to eliminate any effects of the overtraining due to the threshold color ratio value), similar to previous observations of $<$10% attached-type rBC in an urban area (Tokyo) (Moteki



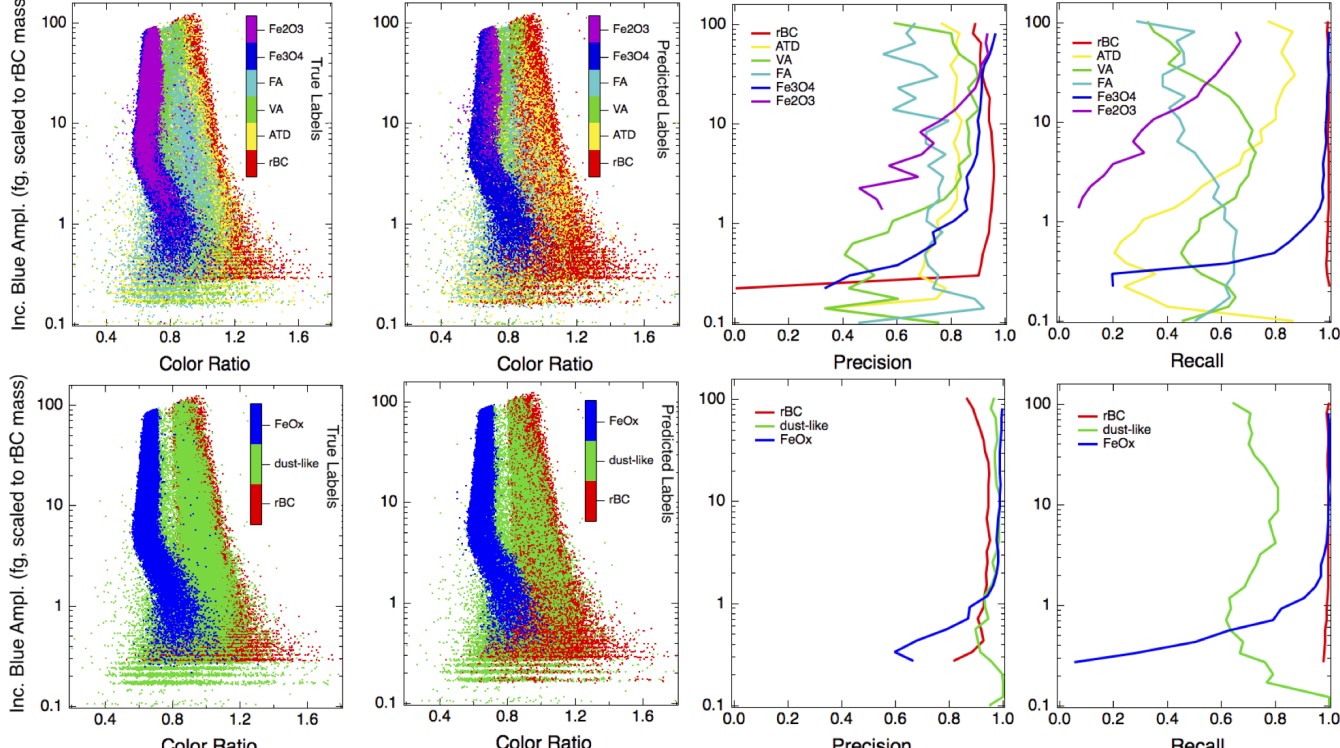

**Figure 8. True and predicted aerosol labels, precision, and recall by incandescent peak height for the 6 class (reduced features), 3 class (reduced features) cases.** The top row shows the performance of the 6 class classification scheme and the bottom row shows the performance of the 3 class classification scheme. The first figure to the left in each row shows the laboratory test data, color coded by its true class label (particles differ between the 6 class and 3 class cases because different subsets of the laboratory data were randomly chosen as test sets in each case). The 2nd figure to the left shows the predictions of the algorithms for the class labels. The 3rd figure shows the precision for each class, binned by incandescent peak height. The 4th figure shows the recall, binned by incandescent peak height for each class.

et al., 2014). This suggests that machine learning could also potentially be used to identify attached-type rBC aerosols over a greater rBC mass range than previously developed algorithms (Moteki et al., 2014).

## 5   Conclusions and Recommendations

We explored the advantages and limitations of using supervised machine learning to classify absorbing aerosols detected via laser-induced incandescence. This paper serves as proof-of-concept that supervised machine learning is a useful technique for analyzing and classifying laser-induced incandescent signals acquired by the SP2. This method improves upon the performance of previous classification methods using only 3 or 4 features derived from the single particle signals, and indicates that the SP2 does provide enough information via laser induced incandescence to identify $FeO_x$ with few misclassifications as other types





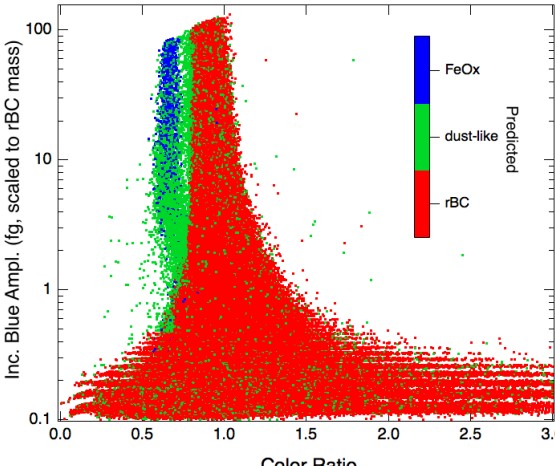

**Figure 9. Application of algorithm to observations in Boulder, CO** Ambient data acquired from a rooftop inlet demonstrates the performance of the 3 class, reduced feature implementation of the random forest algorithm after it has been trained on laboratory data. A clear feature of $FeO_x$ is observed in the ambient data. The larger variety of color ratios at the smaller incandescent peak heights ($<0.5$ fg rBC equivalent mass) than observed in laboratory data is due to the greater prevalence of coatings on the observed rBC, which allows aerosols with a smaller rBC mass to be detected.

of aerosols that the SP2 can detect. In past studies, decision boundaries had been based on inspection of the data rather than statistical considerations, and the method presented here provides a more statistically-consistent method.

In order to use supervised learning algorithms to classify aerosols with an SP2 during aircraft and field observations, we recommend acquiring samples for training data sets with the same instrument, optical configuration, and operating conditions
as the data sets to be processed. Several of the features (in particular the color ratio) demonstrated strong dependence on detector alignment or may be affected by the specific laser power settings. This configuration-dependence leads to a greater incidence of misclassified particles if algorithms trained with data taken with one instrument configuration are applied to data sets attained with another, as the algorithms can be overtrained. This makes the application of the algorithms to aircraft observations more challenging, as changes in pressure and flow rates during sampling may also impact some of these features.
One potential solution is to take a large training data set simulating a number of different alignment configurations, although for simplicity, we have not explored this approach here.

We recommend that the 3 broader class approach be used, as this method provided clear advantages over the 6 class approach. The incandescent onset position of ambient $FeO_x$ observed in East Asia was found to be between that characteristic of $Fe_2O_3$ and $Fe_3O_4$ in pure laboratory samples, suggesting that combustion iron oxide aerosols found in the atmosphere may
be homogeneous internal mixtures of these two iron oxides (Yoshida et al., 2018). This provides additional motivation to use the 3 class classification scheme, as ambient $FeO_x$ may have characteristics on a continuum between pure laboratory samples of $Fe_2O_3$ and $Fe_3O_4$. When applying this method to atmospheric measurements in Boulder, CO, $\sim7\%$ of aerosols in the rBC




mode were quantified as dust-like. These aerosols were associated with a greater incidence of non-volatile particles (that did not evaporate completely in the SP2 laser) than the rBC not identified as dust-like, suggesting that this method may also be useful for identifying rBC attached to other types of aerosols, such as mineral dust. However, because the recall for dust-like aerosols in laboratory samples was only ~70%, there is still significant room for uncertainty in the interpretation of these

aerosols.

Using a random forest ensured good performance and demonstrated that this method in principle works for classifying aerosols detected via laser-induced incandescence; however we have not specifically optimized this method for computational efficiency, and other supervised learning algorithms may offer advantages in this respect. The cross-validation step required the greatest amount of computation time (~1-2 hours for the 3-fold cross-validation grid search using 48 threads), as it required

repeatedly training the model with different options for the hyperparameters. However this step only needs to be performed once; after the hyperparameters have been optimized, the computation time for training the 3 class, reduced features case with ~$6 \times 10^6$ particles was approximately 45 seconds with the optimal set of hyperparameters, and for making predictions on the test set of ~$3 \times 10^6$ particles was <5 seconds. Although we have used a Linux server with multiple processors for this study, we have also deployed this method on a laptop (Macbook Pro with 3 GHz Intel Core i7 processor with 4 cores and 16 GB 1600

MHz DDR3 memory); in this case training time was ~300 seconds and testing time was <5 seconds. Computation time could be reduced by using a smaller training data set, although with some trade-offs in classification accuracy.

Another approach would be to use an unsupervised learning algorithm to classify atmospheric observations, as supervised learning algorithms rely on ambient aerosols having a similar response in the SP2 as laboratory samples. Given the low relative incidence of $FeO_x$ vs. rBC, clustering algorithms that assume consistent cluster size would likely perform poorly, however.

Some unsupervised approaches, such as Hierarchical Agglomerative Clustering analysis, which has previously been used to classify biological aerosols detected via UV light-induced fluorescence (Robinson et al., 2013; Ruske et al., 2017, 2018; Savage and Huffman, 2018), are more appropriate for data sets where cluster size is not expected to be consistent. However, these methods are significantly more computationally intensive than the approach we explored here (e.g. with a time complexity scaling as the square of the number of samples or worse (Müllner, 2011)).

Here we have taken the approach of leveraging previous feature engineering from SP2 incandescent and scattering signals. Since we are using features derived from processing the raw SP2 times series, some features, particularly those associated with the aerosols that do not incandesce with high efficiency and are internally mixed with non-volatile materials (ATD, VA, and FA), may be biased due to detector saturation. One solution to this issue that may improve classification performance for these aerosols would be to use the raw time-resolved single particle signals from the 4 channels directly as features, although

this would be computationally more expensive than the approach taken here. A further adaptation of this method would use representation learning/deep learning to learn features directly from the raw SP2 signals; however, these methods are generally computationally expensive (e.g. often requiring the use of GPU's for parallel processing). These algorithms also have a large number of adjustable parameters that makes their "out-of-the box" application more challenging. We do not consider this approach here but suggest it may be a potentially useful direction for future research.





These results also suggest that machine learning is unlikely to significantly improve the detection of smaller iron oxide aerosols using the SP2, indicating other on-line measurement techniques should be explored for these aerosols. These aerosols have been linked to neurodegenerative diseases such as Alzheimer's, and have even been detected inside the human brain (Maher et al., 2016); improving their atmospheric detection is an important concern for air quality and human health. The worst

classification performance was observed for smaller $FeO_x$ aerosols, although this could in part be due to the size distributions of $FeO_x$ samples in the training data sets. Given that even in the best case scenarios, machine learning algorithms generally do not perform with >98-99% accuracy however, the significantly greater presence of rBC in the atmosphere would likely lead to significant misclassifications of rBC as $FeO_x$ at the smallest sizes even in the best scenarios.

*Code and data availability.* Supervised learning algorithms used in this work are available through the sklearn python package (http://scikit-

10 learn.org/stable/) (Pedregosa et al., 2011). Code used in this study and laboratory data used for training/testing is available upon request from the author.

*Competing interests.* The author declares that she has no conflicts of interest.

*Acknowledgements.* The author would to thank Joshua Schwarz for useful discussion in preparing this paper and carrying out this study. Bernadett Weinzerl is acknowledged for providing the volcanic ash samples and Karl Froyd for providing the samples of coal fly ash.

Douglas Ohlhorst and Richard Tisinai are also acknowledged for their help in accessing the computational resources used to carry out this study. Funding was provided through the NASA Tropospheric Composition Program, the NASA Radiation Sciences Program, and the NASA Upper Atmosphere Research Program.





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
