# Peer review of "Classification of iron oxide aerosols by a single particle soot photometer using supervised machine learning"

_Atmospheric Measurement Techniques, 2019_

## Referee Comment (RC1) · Anonymous Referee #1 · 20 Apr 2019

The author demonstrates for the first time the usefulness of supervised machine learning algorithms for post-processing of the waveform data acquired by the single-particle soot photometer, with particular attention to the classification of iron oxide aerosols. First, the author provides a detailed review of the previous works and clarifies the issues to be solved/mitigated in this work. Second, the author defines the (physical and mathematical) features embedded in the signal waveforms and explain the machine learning algorithm applied to them. Finally, the author shows the suggested algorithm can reduce the chance of misclassification of the iron oxide aerosols than the conventional simpler algorithm. Along with the presentation of the results, the author also fully explains the limitation of the applicability due to a particular selection of laboratory

samples used to train the algorithm. The manuscript is very logically written, and all the figures are easy to understand. Considering the superior quality of discussion and presentation, I can recommend the publication of this work. However, I request minor revisions to improve the readability and influence to a broader audience (including other SP2 users).

Minor comments: Most of the contents in sections 3.3-3.5 look like an overview of "the established theory" of machine learning. If so, the author could shorten these sections (or moved to the supplementary information).

p.2, line 8. nitrogen - > nitrate ?

p.8, line 24. real part -> imaginary part ?

p.18, line 9-10. "retaining the 11 most important features for the 6-class case and the 9 most important for the 3-class case"

Please refer Table3 in this sentence. Otherwise, readers could not follow which features are used here.

p.24, line 6-7. "This method improves upon the performance of previous classification methods using only 3 or 4 features derived from the single particle signals"

Please clarify which features the author mention here.

p.25. line 4-5. "we recommend acquiring samples for training data sets with the same instrument, optical configuration, and operating conditions as the data sets to be processed."

To my opinion, it is better to mention this point as "important requirement" rather than "recommendation".

---

## Referee Comment (RC2) · Anonymous Referee #2 · 29 Apr 2019

The manuscript amt-2019-106 by K. Lamb describes the application of a machine-learning algorithm known as random forests to the data set produced by the NOAA SP2. The work is thorough and the writing is of an excellent calibre. The contribution to the field is significant as this work may significantly influence future SP2 data analyses (hopefully without making them more opaque, which is the inevitable shortcoming of machine learning). I am happy to say that I have only minor requests for information/modified graphs. The manuscript should be published in AMT after the following minor corrections, most of which request language clarification or additional details.

Comments on the abstract:

[Figure]

The abstract specifies that conditional probabilities of each class are provided but then later refers to 'correct identification'. This change of language from probabilistic to absolute identification confused this reviewer on the first read, and the text could be slightly changed to be consistent with a probabilistic perspective (including a definition of what "correct" means in terms of probability... is it a probability of 90%? is it when one class was more than twice as likely as the next? this discussion could also be mentioned in the main text).

Similarly, please explicitly define the "broader class" approach in the abstract. For such technical work, many readers will only read the abstract.

Finally, it should be made clear in the abstract that you are not using an "SP2" but a "modified SP2". This work is not extensible to the standard SP2. Conversely, this work should motivate other SP2 users to modify their SP2s, therefore the modifications must be highlighted.

Minor comments / Requests for information:

I found that the author's decision to include a large amount of detail on the basics of machine learning helpful, and since this is an interdisciplinary journal it is appropriately detailed. However, the author may also consider moving sections of that text to an Appendix to allow the main text of the paper to focus on the essentials.

Page 8, line 16. Why should only the red PMT alignment be sensitive? Is it due to a unique physical configuration of the filter/PMT? Can the author please either speculate or state that this is as surprising to her as it is to the reader?

On page 11 the author mentions that various other machine-learning algorithms were tested with negative results. I think many readers would appreciate more information on these negative results (too often we only report successes). I suggest including a brief appendix (a few paragraphs or table) describing what was done. Surely the author compiled metrics on the different algorithms before deciding to focus on random

forests; this information would be of value to readers who need to know whether their data sets might be significantly different to this one. This would also provide objective support for the manuscript's focus on random forests.

Page 12, it is unclear to me what happened to particles with no valid position-sensitive detector information. Were they rejected?

Please refer to Figure 4 in the legend of Table 2, for the benefit of the non-linear reader. Please also define x4 (post incandescent scattering) more precisely; that is, specify what time interval after scattering was used. Is it defined when incandescence returns to zero? Is it defined for a fixed distance from x8, the position in the laser? Is it possible that this definition influenced the results?

In Section 4.1, several statements such as "most important feature" and "significantly worse classification" were used. It would be helpful if these were quantified numerically, as the reader does not know how to interpret them otherwise. Also please clarify "reduced by approximately 1/3rd", does this mean "reduced by a factor of 0.33" or "...0.7"?

In Section 4.1, a reduced set of training features was justified because "there was a clear break in the relative importance of different features". Presumably the author compiled statistics on prediction accuracy when sequentially removing features, otherwise this statement could not be made. This would be very informative to include as a table or figure.

Page 18 line 29. The private communication with S. Kaspari must definitely be expanded on as it is a very important part of the data interpretation. How did Kaspari prove that the particles were rBC and not dust? Microscopy? Can a quantitative analysis be made?

Section 4.3, the author comments on the low number of dust particles impacting accuracy at small sizes, can this fact be placed in the context of expected FeOx size

distributions? I initially thought it would be insignificant but then a paper on penetration into the brain is cited later.

Page 23 line 26, "misidentifying" based on what? How do you know the true class? It seems like you are somehow convinced that these particles are truly rBC which the algorithm cannot identify – if so, can you please explain why?

Figure 9 legend. Coatings do not allow particles with a smaller rBC mass to be detected (in the incandescence channel). Perhaps the real reason for more smaller particles being detected here is simply more were available (nebulizing fullerene soot produces larger particles than combustion engines). A limit of quantification for the color ratio (eg 0.8 fg) should be defined and discussed.

Figure 9. In my own experience with extremely dense "point plots", I have found that it is impossible to visualize the histogram (or pdf) once the overlap becomes as severe as in this figure. The same problem will occur in Figure 2, but is not misleading (or easy to improve) there. For Figure 9, please add a panel showing the histogram of color ratios for each class, in the region of constant color ratio (>2 fg), or please change to 3 panels of joint PDFs (cumulative count instead of overlapping points), or 3 panels of transparent points.

Page 25 line 5, presumably the author has data to prove this strong dependency? Please show it.

Very minor comments:

Please add abbreviations to Table 1. (Rather than in the legend of Figure 2.)

Contractions such as "it's" are normally discouraged; I leave the details of this to the AMT editing staff.

Page 8, line 12-14. This sentence is grammatically flawed and I can't see what it should be corrected to; please revise.

[Figure]

Figure 3 legend, expand "PS" to position sensitive like in Figure 4.

I would suggest changing "L-II" to "LII" because the latter is an established acronym, and because hyphenated words typically do not retain their hyphens when abbreviated.

Page 11 line 6, change eg to ie.

Page 15 line 8, after "subset" state "discussed below in section ..." for the reader's benefit.

Page 22 line 15, here and later the word "aerosols" starts to creep in to the lexicon, which I find confusing (the author seems to be using "aerosols" as "collection of particles" rather than "suspension of particles in a gas", perhaps "particle ensemble" or "sample set" would be clearer).
* * *

---

## Author Comment (AC1) · 20 Jun 2019

Response to Reviewers

Manuscript: amt-2019-106
Manuscript title: Classification of iron oxide aerosols with a single particle soot photometer using supervised machine learning

The discussion below includes the comments from the reviewer (bold) and my responses to the specific comments (red). Modifications to the manuscript text are given in italics, and line numbers refer to the original document.

Response to Reviewer #1:

**The author demonstrates for the first time the usefulness of supervised machine learning algorithms for post-processing of the waveform data acquired by the single-particle soot photometer, with particular attention to the classification of iron oxide aerosols. First, the author provides a detailed review of the previous works and clarifies the is- sues to be solved/mitigated in this work. Second, the author defines the (physical and mathematical) features embedded in the signal waveforms and explain the machine learning algorithm applied to them. Finally, the author shows the suggested algorithm can reduce the chance of misclassification of the iron oxide aerosols than the conventional simpler algorithm. Along with the presentation of the results, the author also fully explains the limitation of the applicability due to a particular selection of laboratory samples used to train the algorithm. The manuscript is very logically written, and all the figures are easy to understand. Considering the superior quality of discussion and presentation, I can recommend the publication of this work. However, I request minor revisions to improve the readability and influence to a broader audience (including other SP2 users).**

I would like to thank the reviewer for their positive comments and very useful suggestions to improve the clarity and flow of the manuscript. I have addressed their specific comments below.

**Minor comments: Most of the contents in sections 3.3-3.5 look like an overview of "the established theory" of machine learning. If so, the author could shorten these sections (or moved to the supplementary information).**

Thank you for this comment. I have chosen to move the discussion about decision tree classifiers to an Appendix for the interested reader, but have retained the details about training and optimizing the random forest in the main text for readers who may be coming from an interdisciplinary background.

**p.2, line 8. nitrogen - > nitrate?**

Thank for pointing out this typo; this line has been updated.

**p.8, line 24. real part -> imaginary part?**

Thank you for pointing this out. I have updated the text.

**p.18, line 9-10. "retaining the 11 most important features for the 6-class case and the 9 most important for the 3-class case" Please refer Table 3 in this sentence. Otherwise, readers could not follow which features are used here.**

Thanks for pointing this out. I have updated these lines to include the reference to Table 3. I have also added additional clarification to Section 4.1 on the ranking of feature importance.

*p. 18. L. 9-10. We remove the least important features (retaining the 11 most important features for the 6-class case and the 9 most important for the 3-class case; see Table 3, columns 3 and 5 for the subset of features for each case)*

*p. 18, L. 2-3. The relative importance of different features (given in Table 3) is estimated from the fraction of samples in the data set for which the decision pathway is impacted by that feature (Pedregosa et al. 2011).*

**p.24, line 6-7. "This method improves upon the performance of previous classification methods using only 3 or 4 features derived from the single particle signals" Please clarify which features the author mention here.**

These lines have been updated to clarify the features I was referring to were the incandescent peak height, the color ratio, the core scattering, and the post-incandescent scattering amplitudes.

**p.25. line 4-5. "we recommend acquiring samples for training data sets with the same instrument, optical configuration, and operating conditions as the data sets to be processed." To my opinion, it is better to mention this point as "important requirement" rather than "recommendation".**

Thanks for pointing this out. I've adjusted the language of this paragraph to clarify that using an instrument with the same configuration is absolutely necessary to apply the supervised learning algorithm with confidence. I've also added additional details to the supplemental information demonstrating the dependence of the color temperature ratio on the alignment of the detectors in the instrument (Figure S3).

*p.25 L.4-5 In order to use supervised learning algorithms to classify aerosols with an SP2 during aircraft and field observations, it is very important to acquire the samples for training data sets with the same instrument, optical configuration, and operating conditions as the data sets to be processed. Several of the features (in particular the color ratio) demonstrated strong dependence on detector alignment or may be affected by the specific laser power settings (See Supplementary Materials Figure S3 for additional details).*

[Figure]

Figure S3. Influence of detector alignment on the incandescent blue amplitude (mass) to color temperature ratio relationship for fullerene soot samples (a) Mass vs. color ratio for fullerene soot sampled by the NOAA SP2 on three different occasions, with three independent alignments for the blue and red PMT's. The color ratio in each case was normalized to 1.0 for fullerene soot with a mass of 10 fg. Greater variability in color ratio was observed when the detectors are not well-aligned (as in case 2) (b) Normalized histograms of the color ratios for fullerene soot for particles with masses between 2 and 70 fg for the three different optical alignments demonstrate differences in the width of the distributions.

---

## Author Comment (AC2) · 20 Jun 2019

Response to Reviewers

Manuscript: amt-2019-106
Manuscript title: Classification of iron oxide aerosols with a single particle soot photometer using supervised machine learning

The discussion below includes the comments from the reviewer (bold) and my responses to the specific comments (red). Modifications to the manuscript text are given in italics, and line numbers refer to the original document.

Response to Reviewer #2:

**The manuscript amt-2019-106 by K. Lamb describes the application of a machine-learning algorithm known as random forests to the data set produced by the NOAA SP2. The work is thorough and the writing is of an excellent calibre. The contribution to the field is significant as this work may significantly influence future SP2 data analyses (hopefully without making them more opaque, which is the inevitable shortcoming of machine learning). I am happy to say that I have only minor requests for information/modified graphs. The manuscript should be published in AMT after the following minor corrections, most of which request language clarification or additional details.**

I would like to thank the reviewer for their insightful comments and their careful reading of the paper, which has helped to improve the clarity of discussion and added to several key points in the manuscript. I have addressed their specific comments in detail below.

**Comments on the abstract:**

**The abstract specifies that conditional probabilities of each class are provided but then later refers to 'correct identification'. This change of language from probabilistic to absolute identification confused this reviewer on the first read, and the text could be slightly changed to be consistent with a probabilistic perspective (including a definition of what "correct" means in terms of probability... is it a probability of 90%? is it when one class was more than twice as likely as the next? this discussion could also be mentioned in the main text).**

In the sci-kit learn implementation, the predicted class is the one with the highest mean probability (from the ensemble vote of the random forest in this case). I have updated the abstract to better link the predictions for particles of each type (based on the ensemble vote of the random forest for the features associated with each single particle) to the classification/generalization accuracy over the entire class, based on applying the trained algorithm to the test data sets.

*p1., L.12:* *Predictions of the most likely particle class (the one with the highest mean probability) based on applying the trained random forest algorithm to the single-particle features for test data sets comprising examples of each class are compared with the true class for those particles to estimate generalization performance.*

**Similarly, please explicitly define the "broader class" approach in the abstract. For such technical work, many readers will only read the abstract.**

I have specified the three broader classes to clarify what I mean here.

*p1., L.12: ...and one with three broader classes ("rBC", "anthropogenic FeO$_x$", and "dust-like") for particles with similar SP2-responses.*

**Finally, it should be made clear in the abstract that you are not using an "SP2" but a "modified SP2". This work is not extensible to the standard SP2. Conversely, this work should motivate other SP2 users to modify their SP2s, therefore the modifications must be highlighted.**

Thank you for pointing this out. I've updated the abstract to emphasize that a modified SP2 was used in this study (and in previous studies using an SP2 to identify light-absorbing metallic aerosols).

*p1., L.2: SP2s that have been modified to provide greater spectral contrast between their narrow and broad-band incandescent detectors have previously been used to characterize both refractory black carbon (rBC) and light absorbing metallic aerosols, including iron oxides (FeO$_x$)*

**Minor comments / Requests for information:**

**I found that the author's decision to include a large amount of detail on the basics of machine learning helpful, and since this is an interdisciplinary journal it is appropriately detailed. However, the author may also consider moving sections of that text to an Appendix to allow the main text of the paper to focus on the essentials.**

Thank you for this comment. I have moved some of the details (on how decision trees classifiers work) to an Appendix to streamline the discussion in the main text.

**Page 8, line 16. Why should only the red PMT alignment be sensitive? Is it due to a unique physical configuration of the filter/PMT? Can the author please either speculate or state that this is as surprising to her as it is to the reader?**

I've updated this section to emphasize that, while the relative alignment of the two PMT's appears to be important, the addition of the aperture in front of the red PMT makes the color ratio particularly sensitive to the alignment of the red PMT in the NOAA SP2 optical head.

*p.8 L. 16. We also found that the width of the distribution of color ratios for rBC and FeO$_X$ as a function of incandescent peak height strongly depended on 20 the relative alignments of the PMT detectors (See Supplementary Figure S3). Because of the additional aperture in front of the red PMT in the NOAA SP2 optical head, the color ratio was particularly sensitive to the alignment of the red PMT.*

**On page 11 the author mentions that various other machine-learning algorithms were tested with negative results. I think many readers would appreciate more information on these negative results (too often we only report successes). I suggest including a brief appendix (a few paragraphs or table) describing what was done. Surely the author compiled metrics on**

**the different algorithms before deciding to focus on random forests; this information would be of value to readers who need to know whether their data sets might be significantly different to this one. This would also provide objective support for the manuscript's focus on random forests.**

The reviewer rightly points out that exploring other algorithms and approaches for applying machine learning to the SP2 signals would be very useful. As the focus of this work was on demonstrating the potential utility of supervised machine learning for analyzing SP2 signals, I have chosen to de-emphasize the discussion of other algorithms that were initially tested and and move this information to the supplementary information. Using different features or implementations of these algorithms may lead to different outcomes, and I do not want to give the impression that this current research has exhausted the possible approaches. As discussed in Section 3.3, I chose to focus on the random forest because it was straight-forward to implement, it can directly handle multi-class classification problems, and it performed well using the specific features outlined in this work.

*P.11 L. 14. - Here we focus on the application of a random forest algorithm to the SP2 observations (We initially considered other machine learning algorithms, and some additional details are provided in the Supplementary).*

**Page 12, it is unclear to me what happened to particles with no valid position-sensitive detector information. Were they rejected?**

Particles were rejected only in the cases when they did not have valid information for either the incandescent blue peak amplitude or the color ratio. For all the other features, the feature vector for particles with incomplete information were imputed with values during the preprocessing steps, as described in Section 3.2. In particular for the position sensitive detector, the values were chosen to be outside the typical range of values for that feature. I've added additional clarification to the discussion on data preprocessing and also added a reference to Section 3.2 in Section 3.1 to make this clearer to the reader. I've also added Table S1 to the Appendix to show how each feature was preprocessed/imputed in this scheme.

*P. 12, L. 23. Since the application of a machine learning algorithm requires a value for every element in the feature vector, single particle signals that do not have valid values for each of the features given in Table 2 are imputed with dummy values; we discuss the details of this imputation in the next section.*

*P. 12. L. 31. We perform several preprocessing steps to prepare the data for use in the algorithm. (These steps are summarized in Supplementary Information Table S1.)*

**Table S1. Summary of preprocessing steps for each of the features considered in this study**

| Symbol | Feature | Pre-processing | Imputation for particles without valid values |
|--------|---------|----------------|-----------------------------------------------|
| $x_0$ | Blue peak amplitude | Natural Log. | Particles without valid values were rejected |
| $x_1$ | Color ratio | | Particles without valid values were rejected |
| $x_2$ | Core scattering | Natural Log. | 110% of maximum possible value |
| $x_3$ | Total scattering max. | Natural Log. | 110% of maximum possible value |
| $x_4$ | Post incandescent scattering | | Dummy value - low negative value |
| $x_5$ | Evaporation scattering size | Natural Log. | 110% of maximum possible value |
| $x_6$ | Position sensitive wideness | | Dummy value - low negative value |
| $x_7$ | Min. scattering before incandescence | | Dummy value - low negative value |
| $x_8$ | Position sensitive trigger position | | Dummy value - low negative value |
| $x_9$ | Scatter peak location | | Dummy value - low negative value |
| $x_{10}$ | Saturation width | | Dummy value - zero |
| $x_{11}$ | Incandescent start position | | Dummy value - low negative value |
| $x_{12}$ | Evaporation point | | Dummy value - low negative value |
| $x_{13}$ | Incandescent total length | | Dummy value - zero |
| $x_{14}$ | Incandescent used length | | Dummy value - low negative value |
| $x_{15}$ | Light on laser intensity | Natural Log. | 110% of maximum possible value |
| $x_{16}$ | Width fraction from center | | Dummy value - low negative value |

**Please refer to Figure 4 in the legend of Table 2, for the benefit of the non-linear reader. Please also define x4 (post incandescent scattering) more precisely; that is, specify what time interval after scattering was used. Is it defined when incandescence returns to zero? Is it defined for a fixed distance from x8, the position in the laser? Is it possible that this definition influenced the results?**

I have added a reference to Figure 4 in the legend of Table 2, and I have also included additional information in Section 3.1 on how $x_4$ is defined, and cross-referenced this discussion to Section 2 where this feature was discussed in the context of the different incandescent particle types. The post incandescent scattering is defined as the maximum scattering after the incandescence has effectively returned to zero (i.e. is less than some threshold value above the base line for incandescent), and is not referenced to a specific point in the laser. This feature could be impacted by where in the laser beam a particle incandesces, although the value for this feature for particles of the same material should remain consistent.

*P.12 L. 10. Post-incandescent scattering ($x_4$) is defined as the maximum value of the scattering signal after the blue incandescent signal has reached a peak and has returned to the baseline.*

**In Section 4.1, several statements such as "most important feature" and "significantly worse classification" were used. It would be helpful if these were quantified numerically, as the reader does not know how to interpret them otherwise. Also please clarify "reduced by approximately 1/3rd", does this mean "reduced by a factor of 0.33" or "...0.7"?**

I have added additional references to the relative importance of different features as given in Table 3 to make it clear what I am referring to by most important feature. Additionally, I have added Table S2 and Table S3 to the Appendix providing additional information about the precision and recall for the 3 class and 6 classes cases using the different subsets of the features to train the algorithm. I updated Section 4.1 to reference these tables.

**Table S2. Summary of classification accuracy for the 3-class case, using different subsets of feature space.** We provide classification accuracy for the optimized algorithm when using only the $n$ most important features for the 3 class case.

| Class | 17 features | | 9 features | | 5 features | |
|---|---|---|---|---|---|---|
| | Precision | Recall | Precision | Recall | Precision | Recall |
| rBC | 0.94 | 1.00 | 0.93 | 0.99 | 0.84 | 0.99 |
| dust-like | 0.97 | 0.76 | 0.96 | 0.73 | 0.91 | 0.36 |
| $FeO_x$ | 0.99 | 0.99 | 0.99 | 0.99 | 0.98 | 0.99 |

**Table S3. Summary of classification accuracy for the 6-class case, using different subsets of feature space.** We provide classification accuracy for the optimized algorithm when using only the $n$ most important features for the 6 class case.

| Class | 17 features | | 11 features | |
|---|---|---|---|---|
| | Precision | Recall | Precision | Recall |
| rBC | 0.94 | 1.00 | 0.94 | 1.00 |
| ATD | 0.80 | 0.64 | 0.79 | 0.61 |
| VA | 0.81 | 0.66 | 0.81 | 0.66 |
| FA | 0.75 | 0.58 | 0.73 | 0.56 |
| $Fe_3O_4$ | 0.92 | 0.99 | 0.91 | 0.99 |
| $Fe_2O_3$ | 0.87 | 0.50 | 0.87 | 0.50 |

I have clarified p.18 L.17 to read:

*...in the case of the 6 class case, the training time was reduced from 92 to 52 seconds when using 11 rather than 17 features for each sample.*

**In Section 4.1, a reduced set of training features was justified because "there was a clear break in the relative importance of different features". Presumably the author compiled statistics on prediction accuracy when sequentially removing features, other- wise this statement could not be made. This would be very informative to include as a table or figure.**

The importance of different features was based on the ranking of features given by the random forest in the sci-kit learn implementation (as discussed in the first paragraph of Section 4.1), and this ranking motivated which features were retained when training and optimizing the algorithm with a reduced set of features. I have clarified this reference in Section 4.1. As discussed above, I also added additional discussion and Tables S2 and S3 to the Supplementary Materials to demonstrate how the precision and recall is impacted for the 3 class and 6 class cases when running the algorithm with all or a reduced set of features.

*p. 18, L. 2-3. The relative importance of different features (given in Table 3) is estimated from the fraction of samples in the data set for which the decision pathway is impacted by that feature (Pedregosa et al. 2011).*

**Page 18 line 29. The private communication with S. Kaspari must definitely be expanded on as it is a very important part of the data interpretation. How did Kaspari prove that the particles were rBC and not dust? Microscopy? Can a quantitative analysis be made?**

The SP2 color ratio for these rBC measured in ATD samples were consistent with rBC. After the ATD samples were heat-treated, significantly fewer refractory aerosols were detected. I have added these details to the discussion in Section 4.2 and also added a reference to the color ratio histograms for the laboratory samples in Supplementary Figure S1. The histograms of the color ratios for the laboratory samples indicates that the fly ash sample in particular demonstrates clear evidence of two different color ratio modes, with the mode at higher color temperature ratios consistent with the rBC and fullerene soot samples.

*p. 18. L28-29. Previous work has noted that there is a small fraction of rBC present in laboratory samples of ATD (S. Kaspari, private communication), which likely contributes to the high rate of errors between ATD and rBC. (Color ratios of particles detected by the SP2 in these ATD samples were consistent with rBC, and were subsequently removed after heat treating the samples - S. Kaspari, private communication). A significant fraction of the color ratios for the incandescent aerosols detected in the FA laboratory samples also demonstrated a color ratio distribution more consistent with rBC, suggesting a fraction of the incandescent aerosols detected in fly ash may also be rBC (See Supplementary Figure S1).*

[Figure]

**Figure S1. Normalized histograms for the color ratios of laboratory samples and ambient aerosols shown in Figure 2 in the manuscript.**

**Section 4.3, the author comments on the low number of dust particles impacting accuracy at small sizes, can this fact be placed in the context of expected FeOx size distributions? I initially thought it would be insignificant but then a paper on penetration into the brain is cited later.**

I would like to thank the reviewer for this useful comment; the reviewer rightly points out that ambient $FeO_x$ size distributions may differ from the laboratory samples, which could impact the accuracy of the algorithm in identifying ambient particles at the smaller sizes. Several recent papers have provided $FeO_x$ size distributions (as observed with a modified SP2 in East Asia) [Yoshida et al. 2016; Moteki et al. 2017; Yoshida et al. 2018]. I have added Figure S2 to the appendix, demonstrating how the laboratory sample size distributions compare with previous ambient observations. I have also chosen to expand the discussion in Section 5 to include a reference to the expected size distributions for $FeO_x$ in ambient samples. Based on the size distributions of the laboratory samples, I have also updated the discussion related to the smallest $FeO_x$ aerosols (that may be relevant for health/air quality) to suggest caution must be used to choose an appropriate training data set to acquire observations of particles with smaller volume equivalent diameters.

[Figure]

**Figure S2. Normalized dM/dLogD and dN/dLogD size distributions for the iron oxide samples observed by the SP2, compared with ambient observations of FeO$_x$ in East Asia**

*p. 27. L 1-2. Several recent observations of the size distributions of ambient FeO$_x$ in East Asia have indicated a significant number fraction of FeO$_x$ at smaller sizes (<300 nm) (Moteki et al. 2017, Yoshida et al. 2018); however, the nebulized samples of Fe$_2$O$_3$ and Fe$_3$O$_4$ in the laboratory data sets were predominantly between 350-1200 nm volume equivalent diameter (See Supplementary Figure S2). These results indicate that particular care needs to be taken when acquiring a training data set appropriate for classifying smaller iron oxide aerosols.*

**Page 23 line 26, "misidentifying" based on what? How do you know the true class? It seems like you are somehow convinced that these particles are truly rBC which the algorithm cannot identify – if so, can you please explain why?**

I am basing this on the color ratio modes, and the abrupt transition from identifying mainly rBC aerosols to mainly dust-like aerosols for larger particles (>5 fg rBC equiv. mass.) with color ratios less than 0.8 that are on the shoulder of the rBC mode. To better emphasize the transition that I'm referring to, I have added histograms of the color ratios for the more massive particles to Figure 9. I have also added a comparison of the histograms of the color ratios for the laboratory samples to ambient aerosols in Supplementary Figure S1, which demonstrate that there is a greater prevalence of slightly lower color ratios for the ambient rBC when compared with the fullerene soot samples. I have updated the discussion in Section 4.4 to address this point.

*p.23. L.25-29. The algorithm also identifies a significant fraction of dust-like aerosols, both at cooler color temperature ratios, and mixed into the population of aerosols in the rBC mode. At the larger masses for the rBC color ratio mode, the algorithm does appear to be misidentifying a fraction of the rBC with cooler color temperature ratios as dust-like aerosols (as all particles below ~0.8 and with incandescent blue amplitudes > 5 fg rBC equivalent incandescence on the shoulder of the rBC mode are identified as dust-like in Figure 9). This mis-identification is likely due to the differences between the SP2 response to ambient rBC vs. fullerene soot, as ambient rBC has a greater prevalence of particles with a lower color temperature ratio than fullerene soot (See Supplementary Figure S1).*

**Figure 9 legend. Coatings do not allow particles with a smaller rBC mass to be detected (in the incandescence channel). Perhaps the real reason for more smaller particles being detected here is simply more were available (nebulizing fullerene soot produces larger particles than combustion engines). A limit of quantification for the color ratio (eg 0.8 fg) should be defined and discussed.**

Thank you for pointing this out. I've updated the discussion in the caption of Figure 9 to discuss the difference in size between the nebulized fullerene soot aerosols and the greater prevalence of small rBC seen in an urban environment.

*Figure 9 caption. The larger variety of color ratios at the smaller incandescent peak heights (<0.5 fg rBC equivalent mass) than observed in laboratory data is due to the greater prevalence of small rBC aerosols in the urban environment than in the nebulized fullerene soot samples.*

**Figure 9. In my own experience with extremely dense "point plots", I have found that it is impossible to visualize the histogram (or pdf) once the overlap becomes as severe as in this figure. The same problem will occur in Figure 2, but is not misleading (or easy to improve) there. For Figure 9, please add a panel showing the histogram of color ratios for each class, in the region of constant color ratio (>2 fg), or please change to 3 panels of joint PDFs (cumulative count instead of overlapping points), or 3 panels of transparent points.**

For Figure 9, I have added a histogram showing the color ratios for the three different classes, both for all the particles identified with each class, and for only larger particles.

[Figure]

Figure 9. Application of algorithm to observations in Boulder, CO (a) Ambient data acquired from a rooftop inlet demonstrates the performance of the 3 class, reduced feature implementation of the random forest algorithm after it has been trained on laboratory data. A clear feature of $FeO_x$ is observed in the ambient data. The larger variety of color ratios at the smaller incandescent peak heights (< 0.5 fg rBC equivalent mass) than observed in laboratory data is due to the greater prevalence of small rBC aerosols in the urban environment than in the nebulized fullerene soot samples. (b) Histograms for the color ratios of the particles identified to belong to each of the 3 classes are shown, both for the entire population identified and also only for particles with larger incandescent blue amplitudes.

**Page 25 line 5, presumably the author has data to prove this strong dependency? Please show it.**

I have added some additional details in the Supplementary (Figure S3) demonstrating the color ratio dependence on detector alignment for fullerene soot samples for three different optical alignments of the red and blue detectors in the NOAA SP2.

[Figure]

Figure S3. Influence of detector alignment on the incandescent blue amplitude (mass) to color temperature ratio relationship for fullerene soot samples (a) Mass vs. color ratio for fullerene soot sampled by the NOAA SP2 on three different occasions, with three independent alignments for the blue and red PMT's. The color ratio in each case was normalized to 1.0 for fullerene soot with a mass of 10 fg. Greater variability in color ratio was observed when the detectors are not well-aligned (as in case 2) (b) Normalized histograms of the color ratios for fullerene soot for particles with masses between 2 and 70 fg for the three different optical alignments demonstrate differences in the width of the distributions.

**Very minor comments:**
**Please add abbreviations to Table 1. (Rather than in the legend of Figure 2.)**

Done.

**Contractions such as "it's" are normally discouraged; I leave the details of this to the AMT editing staff.**

I have removed contractions.

**Page 8, line 12-14. This sentence is grammatically flawed and I can't see what it should be corrected to; please revise.**

I have updated these lines to improve the clarity of discussion.

**Figure 3 legend, expand "PS" to position sensitive like in Figure 4.**

I have updated the legends in both subfigures.

**I would suggest changing "L-II" to "LII" because the latter is an established acronym, and because hyphenated words typically do not retain their hyphens when abbreviated.**

Thanks for pointing this out; I have updated the acronyms to be more consistent with previous literature.

**Page 11 line 6, change eg to ie.**

I have updated this line.

**Page 15 line 8, after "subset" state "discussed below in section …" for the reader's benefit.**

I have added this reference.

**Page 22 line 15, here and later the word "aerosols" starts to creep in to the lexicon, which I find confusing (the author seems to be using "aerosols" as "collection of particles" rather than "suspension of particles in a gas", perhaps "particle ensemble" or "sample set" would be clearer).**

Thanks for this clarification. I have updated the language here and also throughout the rest of this section to improve the clarity of discussion.